# Bacteria use exogenous peptidoglycan as a danger signal to trigger biofilm formation

**Sanika Vaidya** [1,6], **Dibya Saha** [2,8], **Daniel K. H. Rode** [2,8], **Gabriel Torrens** [3], **Mads F. Hansen** [1,7], **Praveen K. Singh**[1], **Eric Jelli**[1,4], **Kazuki Nosho** [2], **Hannah Jeckel** [2], **Stephan Göttig** [5], **Felipe Cava** [3] **& Knut Drescher** [2] ✉

For any organism, survival is enhanced by the ability to sense and respond to threats in advance. For bacteria, danger sensing among kin cells has been observed, but the presence or impacts of general danger signals are poorly understood. Here we show that different bacterial species use exogenous peptidoglycan fragments, which are released by nearby kin or non-kin cell lysis, as a general danger signal. Using microscopy and gene expression profiling of *Vibrio cholerae*, we find that even brief signal exposure results in a regulatory response that causes three-dimensional biofilm formation, which protects cells from a broad range of stresses, including bacteriophage predation. A diverse set of species (*Pseudomonas aeruginosa*, *Acinetobacter baumannii*, *Staphylococcus aureus*, *Enterococcus faecalis*) also respond to exogenous peptidoglycan by forming biofilms. As peptidoglycan from different Gram-negative and Gram-positive species triggered three-dimensional biofilm formation, we propose that this danger signal and danger response are conserved among bacteria.

When bacteria experience stresses that threaten their survival or inhibit their growth, protective adaptations occur on different timescales. After exposing cells to environmental stresses for several generations, genotypes with protective traits are selected. On shorter timescales, immediately after exposing cells to stress, the bacteria often sense the stress and regulate a response to survive and replicate[1]. It would be most advantageous for bacteria, however, if the cells could sense and respond to a threat even before they are directly exposed. Such danger sensing has been widely investigated for human immune cells, which can sense bacteria, fungi or tissue damage through many signals that range in information content from very specific to very general danger. Danger sensing has also recently been reported for bacteria to take place between kin cells[2–4], which led us to speculate that danger sensing could be more widely used among bacteria and that there might be conserved danger signals.

While investigating the response of *Vibrio cholerae* to bacteriophage (phage) predation, which is of great importance for the outbreaks of the cholera disease[5] and likely impacts the severity of the disease[6–8], we discovered a general bacterial danger signal. This danger signal carries information not only about the presence of phages but also about any adverse condition causing lysis of Gram-negative or Gram-positive bacteria in the vicinity. Biofilms have been shown to protect bacterial cells from many environmental threats, including phage predation[9–12]. We observed a response to the general danger signal which is conserved across many bacterial species: the formation of protective biofilms.

## Results

### *V. cholerae* forms biofilms during phage exposure

To investigate and directly observe the response of *V. cholerae* to phages, we inoculated *V. cholerae* wild type (WT) cells in microfluidic

[1]Max Planck Institute for Terrestrial Microbiology, Marburg, Germany. [2]Biozentrum, University of Basel, Basel, Switzerland. [3]The Laboratory for Molecular Infection Medicine Sweden (MIMS), Umeå Center for Microbial Research (UCMR), Science for Life Laboratory (SciLifeLab), Department of Molecular Biology, Umeå University, Umeå, Sweden. [4]Department of Physics, Philipps-Universität Marburg, Marburg, Germany. [5]Institute of Medical Microbiology and Infection Control, University Hospital Frankfurt, Frankfurt am Main, Germany. [6]Present address: Antimicrobial Discovery Center, Department of Biology, Northeastern University, Boston, MA, USA. [7]Present address: Section of Microbiology, University of Copenhagen, Copenhagen, Denmark. [8]These authors contributed equally: Dibya Saha, Daniel K. H. Rode. ✉e-mail: knut.drescher@unibas.ch

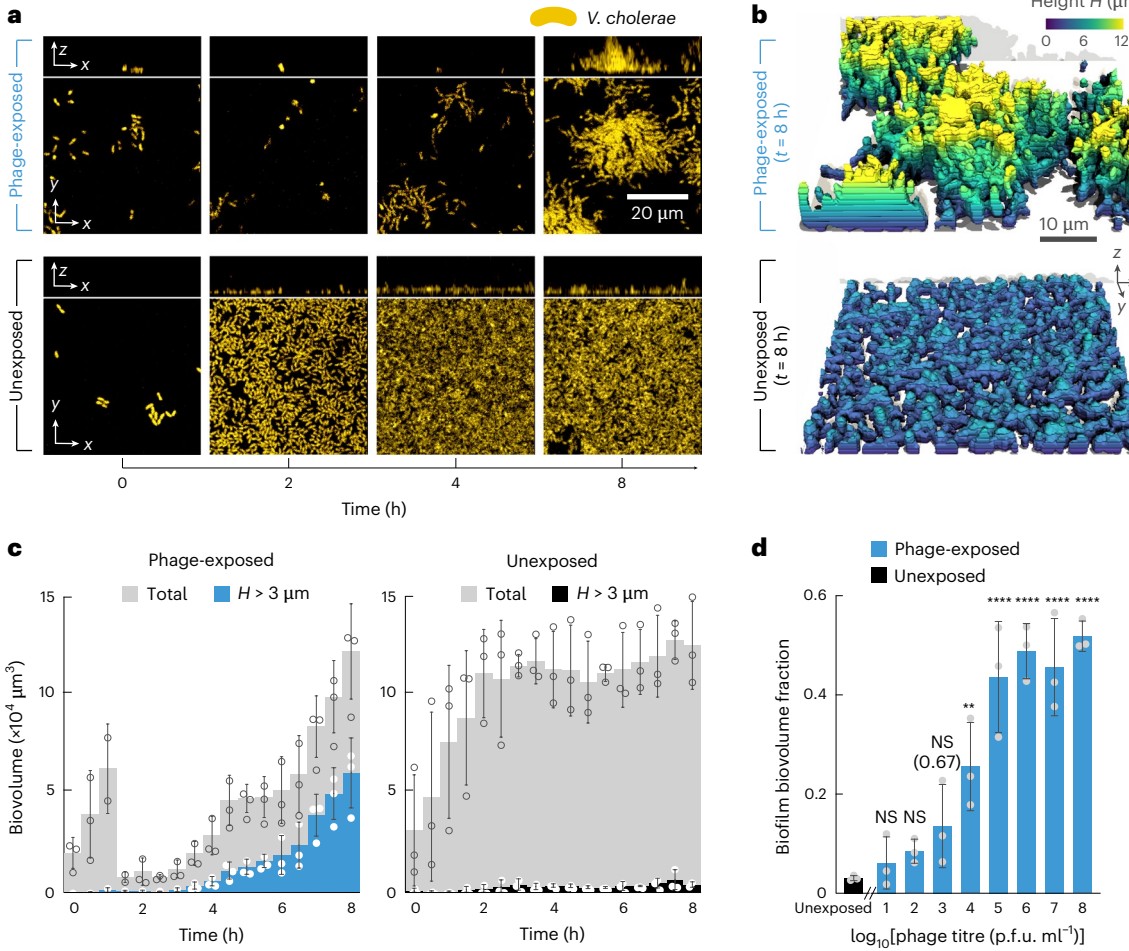

**Fig. 1 | *V. cholerae* forms 3D biofilms following phage exposure. a**, Confocal image time series of *V. cholerae* (shown in yellow, constitutively expressing *sfGFP*) grown in a flow chamber. At time $t = 0$ h, cells were exposed to a continuous flow of Vibriophage N4 (phage-exposed, $10^6$ p.f.u. ml$^{-1}$, top row of images) or medium without phages (unexposed, bottom row). Bacterial cells that are continuously exposed to phages show an initial decrease in total biovolume due to phage infection, followed by growth as 3D biofilms. In the absence of phage exposure, cells grow as a surface-covering 2D lawn without 3D structure. **b**, Rendered images of phage-exposed (top) and unexposed (bottom) *V. cholerae* cells at $t = 8$ h. Cells are coloured according to their height $H$ above the bottom surface of the microfluidic chamber. **c**, Quantification of the total bacterial biovolume (grey) and biovolume with height $H > 3$ μm over time, when the cells are exposed (blue, left panel) or unexposed (black, right panel) to Vibriophage N4 ($10^6$ p.f.u. ml$^{-1}$). Biovolume is quantified using BiofilmQ as the volume (μm$^3$) occupied by fluorescent bacterial cells. **d**, 3D biofilm formation increases with increasing phage titre. 3D biofilm formation is quantified here as the 'biofilm biovolume fraction', measured after 8 h. The 'biofilm biovolume fraction' is defined as the biovolume with height $H > 3$ μm, divided by the total biovolume. In panels **c** and **d**, bars are mean values of $n = 3$ independent biological replicates, circles indicate individual measurements and error bars indicate the standard deviation. Statistical significances in panel **d** were calculated relative to the unexposed condition, using a one-way ANOVA (analysis of variance) with Bonferroni's correction; NS, not significant ($P > 0.99$, except for unexposed versus phage titre $10^3$ where $P = 0.67$); **$P = 0.0081$; ****$P < 0.0001$.

chambers, flushed out non-adherent cells and then exposed them to a continuous flow of fresh Luria–Bertani (LB) medium at 37 °C containing Vibriophage N4 (a T7-like lytic phage[13,14]) at $10^6$ p.f.u. ml$^{-1}$ (plaque-forming units per millilitre), corresponding to an initial multiplicity of infection (MOI) = 1. Phage infection dynamics for different media and temperatures are shown in Extended Data Fig. 1. Using confocal microscopy, we observed that many cells in flow channels that were exposed to phages lysed quickly, but the survivors continued to grow and formed three-dimensional (3D) biofilm colonies within a few hours (Fig. 1a,b). By contrast, cells that were not exposed to phages only grew as a two-dimensional (2D) cell monolayer on the surface of the channel, without forming 3D structures (Fig. 1a,b). To quantitatively distinguish 2D bacterial monolayers from 3D biofilm structures, we used computational image analysis[15] to measure the entire bacterial biovolume in the channel and the biovolume with a height of more than 3 μm above the glass substrate of the flow channel. Bacterial cells exposed to phages showed an increase in the biovolume above 3 μm from the substrate, whereas in the absence of phage exposure the growth of bacterial cells

was limited to less than 3 μm in height, effectively corresponding to a 2D bacterial monolayer (Fig. 1c).

It has been observed that when bacteria are exposed to phages for extended periods[16–18], mutants with increased biofilm formation are selected, because cells inside biofilms are often protected from phages due to limited mobility of phages into the biofilm and due to the obstruction of phage binding sites by the extracellular biofilm matrix[10,19–22]. Control experiments in which the biofilms that had formed after 8 h of phage exposure were broken up into individual cells showed that the bacterial population was as sensitive to phage exposure as cells that had not previously been exposed to phages (Extended Data Fig. 2a,b). Using the crystal violet assay for quantifying the biofilm formation of the cells obtained from biofilms formed after 8 h of phage exposure, we observed that these cells showed no significant difference in biofilm formation capability compared with the WT (Extended Data Fig. 2c). Together, these control experiments indicate that phage-resistant mutants and matrix hyper-producer mutants have not arisen or did not become a substantial fraction of the

population during this relatively brief exposure to phages, suggesting that the increased 3D biofilm production during a few hours of phage exposure might be a regulatory response.

Phage-induced lysis causes a reduction of the cell density (Fig. 1a,c), and it has previously been reported that biofilm extracellular matrix production can be upregulated at low cell density using the quorum sensing regulatory circuit in *V. cholerae*[23]. However, when we performed a systematic reduction of the bacterial seeding density in the channel, we observed that this was not sufficient for inducing 3D biofilm growth (Extended Data Fig. 3), indicating that quorum sensing is unlikely to cause the 3D biofilm growth during phage exposure. It is worth noting that measurements of the fraction of the bacterial biovolume in the flow channel that is bound within 3D biofilms (termed 'biofilm biovolume fraction' in Fig. 1d) for different phage concentrations resulted in a standard dose–response curve that saturates with increasing phage concentration (Fig. 1d). These findings led us to the hypothesis that the cells might sense the presence of phages and respond by forming 3D biofilms.

### Phage-free cell lysate induces biofilm formation

To test the hypothesis that *V. cholerae* can sense the presence of extracellular phages, we investigated how the cells might achieve this. Measurements of biofilm formation during exposure to heat-inactivated phages[24] showed that these inactive virions did not induce 3D biofilm formation (Fig. 2a), suggesting that phage infection was required to induce 3D biofilm formation. To test whether not only phage infection but also phage-induced lysis was required for inducing biofilm formation, we measured the response of *V. cholerae* mutants lacking the *trxA* gene (*vc0306*) to active phages. TrxA is a thioredoxin that confers processivity to the T7 phage DNA polymerase[25,26], which is known to be important for T7 phage replication and lysis of the host. Phage adsorption onto the cells was identical for the WT and Δ*trxA* mutant (Extended Data Fig. 4a). However, the Δ*trxA* mutant was able to grow during phage exposure (Fig. 2b and Extended Data Fig. 4b) because it underwent considerably less lysis during phage exposure, as revealed by efficiency of plating (EOP) assays (Extended Data Fig. 4c). It is worth noting that the Δ*trxA* mutant did not form biofilms in response to phage exposure (Fig. 2c), suggesting that cells did not respond to the phage infection directly but that cellular lysis was important for inducing biofilm formation of the non-lysed cells. To test whether cell lysis was required for inducing biofilm formation, we generated a phage-free cell lysate by sonication and observed that both the WT and the Δ*trxA* mutant formed 3D biofilms during exposure to lysate (Fig. 2d). Measurements of the biofilm growth dynamics during lysate exposure showed that lysate induced 3D biofilm growth already within 3 h, and the resulting biofilm morphology was identical to biofilms resulting from phage exposure (Fig. 2e). The faster 3D biofilm formation during lysate exposure (Fig. 2e), compared with phage exposure (Fig. 1c), could arise from the initial delay in generating the first cell lysis following phage exposure and the continuous reduction of biomass due to the ongoing lysis of phage-sensitive cells in the population. In addition, exposure to lysate slightly increased the bacterial growth rate (Extended Data Fig. 5a). Lysate obtained from higher cell densities also showed a higher capacity for inducing biofilm formation (Fig. 2f).

How does exposure to lysate cause the formation of 3D biofilms? It is conceivable that lysate could promote biofilm formation of *V. cholerae* cells by depositing a compound on the surface of the microfluidic channel that helps cells to attach to the surface. However, control experiments in which the surface of the flow channels was pre-treated with lysate showed that a lysate surface coating did not significantly induce biofilm formation (Extended Data Fig. 6a). Alternatively, components of the lysate could become part of the extracellular matrix to promote biofilm formation, analogous to a process that has been observed in *Pseudomonas aeruginosa* and *Shewanella oneidensis*,

where the DNA released by cellular lysis becomes an important component of the extracellular matrix during biofilm development[27,28]. However, the exposure of *V. cholerae* to different concentrations of extracellular DNA that was purified from the lysate did not result in biofilm formation (Extended Data Fig. 6b). Furthermore, experiments in which isolated *V. cholerae* cells were only exposed to lysate for 10 min, before switching back to media that did not contain any lysate, showed that this very brief exposure triggered 3D biofilm formation to the same level as continuous lysate exposure (Extended Data Fig. 6c). This finding is incompatible with the hypothesis that a component of the lysate becomes an important part of the extracellular matrix during 3D biofilm growth, because the cells only grew into 3D biofilms approximately 1 h after the lysate was removed from the medium. We therefore explored the possibility that cells sense one or more components of the lysate and mount a regulated response that involves the growth into 3D biofilms.

### Peptidoglycan in cell lysate induces biofilm formation

To identify whether *V. cholerae* senses a particular component of the *V. cholerae* cell lysate, and what class of compound this could be, we initially tested whether *V. cholerae* also forms 3D biofilms in response to lysates from other Gram-negative species (*Escherichia coli* and *P. aeruginosa*) and Gram-positive species (*Bacillus subtilis* and *Staphylococcus aureus*). All cell lysates were obtained by sonication of $10^{10}$ cells per ml. It is worth noting that *V. cholerae* formed 3D biofilms following exposure to cell lysate irrespective of which bacterial species was used to generate the lysate (Fig. 3a), indicating that the biofilm-inducing signal is not particular to any given species. Treating the *V. cholerae* lysate with heat, DNase, RNase or proteinase K did not significantly diminish its capacity to induce 3D biofilm formation of *V. cholerae* (Extended Data Fig. 7). Finally, we investigated the size of the active component in the lysate using filters of different pore size. These measurements showed that the biofilm-inducing molecules have a wide range of sizes, but they are larger than 3 kDa (Fig. 3b). Together, the results from the different treatments of the lysate show that proteins or nucleic acids in the lysate do not induce 3D biofilm formation and that the active components of the lysate have a range of different sizes and are present in widely different bacterial species.

To test whether cell wall fragments could be a biofilm-inducing signal, we generated cell lysate from spheroplasts, which lack a cell wall due to lysozyme treatment[29], and we separately purified the cell wall from *V. cholerae* WT cells. While exposure to purified cell wall fragments was capable of inducing 3D biofilm formation to similar levels as the cell lysate, exposure to spheroplast lysate resulted in significantly less biofilm formation (Fig. 3c). These results show that exogenously supplied *V. cholerae* cell wall fragments are capable of inducing 3D biofilm formation and that there are potentially other components of the cell lysate that also induce biofilm formation. These findings were corroborated by experiments in which *V. cholerae* was exposed to 300 μg ml$^{-1}$ purified peptidoglycan (PG) (corresponding to a lysate of ~$10^{10}$ cells per ml), which can be conveniently obtained commercially (of *B. subtilis* origin). Exposure to purified PG resulted in 3D biofilm growth dynamics and biofilm morphology (Fig. 3d) that are similar to biofilms arising from exposure to cell lysate (Fig. 2e). PG induced 3D biofilm formation even when experiments were performed in different media and temperatures (Extended Data Fig. 8). The presence of PG neither increased nor decreased the growth rate of *V. cholerae* (Extended Data Fig. 5b). Furthermore, exposing *V. cholerae* to a higher concentration of PG resulted in a larger amount of 3D biofilm biovolume in a dose-dependent manner (Fig. 3e). Treatment of PG with distinct PG-degrading enzymes narrowed down the structural nature of the biofilm-inducing component (Fig. 3f): Digestion of purified *V. cholerae* PG with lysozyme or amidase resulted in a loss of the biofilm induction capacity, which confirms that exposure to exogenous PG is sufficient for inducing 3D biofilm formation in *V. cholerae*. Digests of PG with

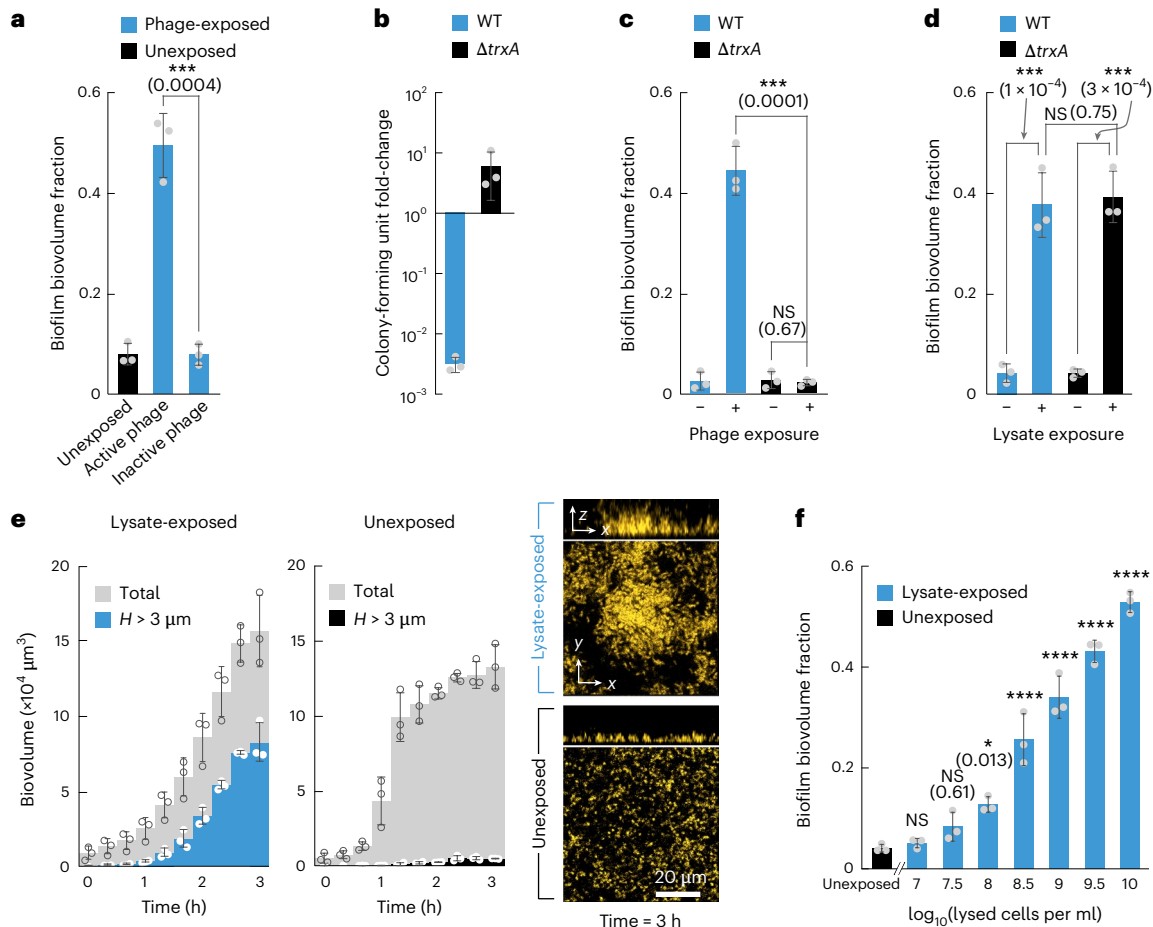

**Fig. 2 | *V. cholerae* 3D biofilm formation is induced by cell lysate. a**, *V. cholerae* forms biofilms during exposure to active Vibriophage N4 ($10^6$ p.f.u. ml$^{-1}$; see also Fig. 1d) but not when phages were inactivated by heat treatment (65 °C for 15 min). Biofilm formation is quantified as the 3D biofilm biovolume fraction (biovolume with height $H > 3$ µm divided by the total biovolume). Measurements were performed after 8 h of exposure to active or inactive virions. **b**, Following 60 min of phage exposure in liquid shaking culture with MOI = 1, the bacterial colony-forming unit count dropped substantially for the WT due to phage-induced lysis, whereas the Δ*trxA* mutant was unaffected by phage exposure. **c**, The Δ*trxA* mutant did not form 3D biofilms after 8 h of phage exposure ($10^6$ p.f.u. ml$^{-1}$). The WT or Δ*trxA* mutant did not form 3D biofilms without phage exposure. **d**, Both the WT and Δ*trxA* mutant formed biofilms when exposed to a lysate obtained by sonication of WT *V. cholerae* cells ($10^{10}$ lysed cells per ml) for 3 h. **e**, Quantification of the total biovolume (grey) and biovolume with height

$H > 3$ µm, measured in the presence (blue, left panel) or absence (black, right panel) of a lysate obtained by sonication of WT *V. cholerae* cells ($10^{10}$ lysed cells per ml). Confocal microscopy images show *V. cholerae* cells expressing *sfGFP* (displayed in yellow) after 3 h in the lysate-exposed or unexposed condition. **f**, Exposure to increasing concentrations of *V. cholerae* lysate solutions (obtained by sonication) resulted in a higher 3D biofilm biovolume fraction. In all panels (**a–f**), bars are mean values of $n = 3$ independent biological replicates, circles indicate individual measurements and error bars indicate the standard deviation. Statistical significances were calculated between the groups indicated by black lines in panels **a–d** using a two-sided Student's *t*-test or relative to the unexposed condition in panel **f** using a one-way ANOVA with Bonferroni's correction. Statistical results are given as exact *P* values in brackets in the graphs, or indicated using the following: NS, not significant ($P > 0.99$, unless specified otherwise in the figure), ****$P < 0.0001$.

endopeptidase or lytic transglycosylase retain some biofilm induction capacity but with a significant reduction compared with undigested PG (Fig. 3f). Considering the most abundant muropeptides produced in both the lytic transglycosylase and endopeptidase digestions[30], we speculate that tetrapeptide anhydro-disaccharides (either free or as part of uncrosslinked PG chains) may be the components of PG that cause biofilm induction.

To test whether exogenously added PG is a signal or whether it is used as a matrix material to promote the construction of the biofilm, we performed experiments in which we exposed *V. cholerae* to PG for only 5 min and observed that this brief period of exposure was sufficient for inducing 3D biofilm growth (Fig. 3g). These results confirm the hypothesis that exogenous PG is sensed by *V. cholerae*. These results also show that the daughter cells of the cells that were exposed to PG for 5 min continue to form biofilms, even though they did not directly get exposed to PG, suggesting that the switch to the biofilm formation phenotype is passed on to daughter cells.

## Exogenous PG induces c-di-GMP and matrix production

As biofilm formation is triggered by sensing exogenous PG, this response likely involves transcriptional regulation. To characterize the regulatory response, we measured the transcriptomes of *V. cholerae* cells in flow chambers following 10 min of exposure to purified PG in LB medium or 10 min of exposure to LB medium without PG, using RNA sequencing (RNA-seq) (Fig. 4a). By comparing these transcriptomes, we identified 325 genes that were upregulated and 70 genes that were downregulated (fold changes >2 or <−2, FDR-adjusted $P < 0.05$) in response to exogenous PG. The upregulated genes can be classified into several functional categories (Fig. 4b) and are listed in Source Data Fig. 4. The upregulated biofilm matrix-related genes are the *vps*-I and *vps*-II gene clusters, which encode the biosynthesis pathway for the production of *Vibrio* polysaccharide (VPS), an essential component of 3D *V. cholerae* biofilms[31]. Genes coding for several virulence factors, such as the *tcp* and *mak* operons, as well as *hlyA* and *hapA*, and the CBASS phage defence system were also upregulated after 10 min exposure to PG.

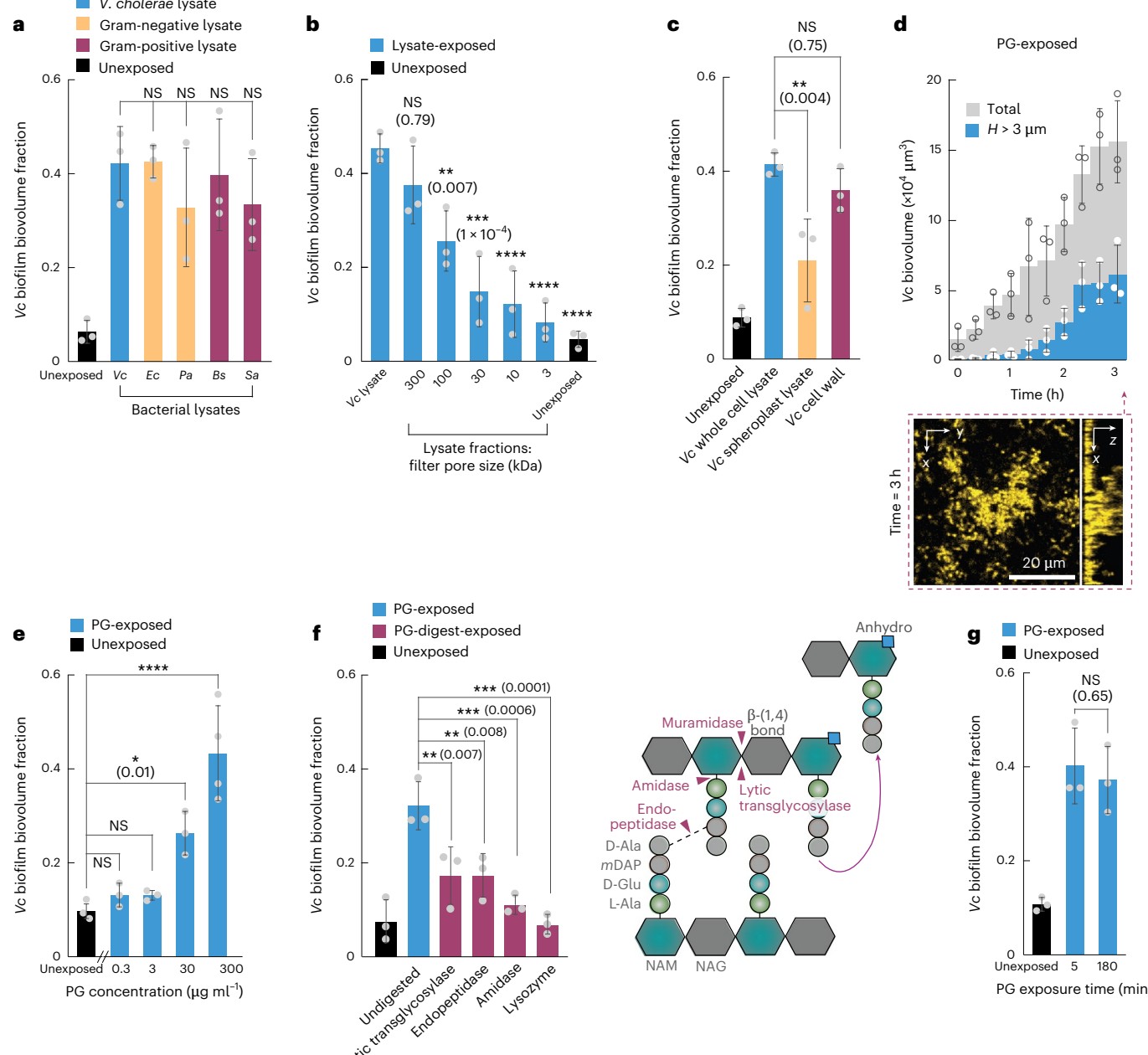

**Fig. 3 | Exogenous PG triggers the *V. cholerae* 3D biofilm formation program.** **a**, *V. cholerae* (*Vc*) cells grew into 3D biofilms when exposed to lysate ($10^{10}$ lysed cells per ml, obtained by sonication) of *V. cholerae* cells (blue bar; see also Fig. 2f) or lysates of other Gram-negative species (yellow bars: *Ec*, *E. coli*; *Pa*, *P. aeruginosa*) or lysates of Gram-positive species (purple bars: *Bs*, *B. subtilis*; *Sa*, *S. aureus*). Biofilm formation was quantified as the 3D biofilm biovolume fraction after 3 h of exposure to lysate (or unexposed control). **b**, Different fractions of a *V. cholerae* lysate, obtained by filtration with different pore sizes (3–300 kDa), showed reduced biofilm induction capacity for smaller filter pore sizes. The lysate was obtained by sonicating $10^{10}$ *Vc* cells per ml, followed by sterilization using a 0.22 µm filter, followed by fractionation with filters of different pore sizes. **c**, Comparison of the biofilm induction capacity of different *V. cholerae* lysates: lysate of WT whole cells (blue, similar to **a** and Fig. 2f), lysate of cells that lacked a cell wall (spheroplasts, yellow) or cell wall fragments purified from a lysate of WT whole cells (purple), relative to the unexposed condition (black). **d**, Exogenously added pure PG (300 µg ml$^{-1}$) induced 3D biofilm formation of *V. cholerae*. Confocal microscopy image shows *V. cholerae* cells (yellow, constitutively expressing *sfGFP*) exposed to PG for 3 h.

**e**, *V. cholerae* biofilm formation after 3 h of PG exposure increased with increasing concentration of PG. Biofilm formation was quantified as the fraction of 3D biofilm biovolume with height $H > 3$ µm. We estimate that a PG concentration of 300 µg ml$^{-1}$, solubilized by sonication, approximately corresponds to the PG concentration in lysate of $10^{10}$ lysed cells per ml. **f**, *V. cholerae* WT cells were exposed to purified *V. cholerae* PG (300 µg ml$^{-1}$ in LB) which was either undigested or treated with enzymes that cleave specific bonds in PG. The scheme illustrates which bonds are cleaved by each enzyme. **g**, Exposure of *V. cholerae* WT cells to PG (300 µg ml$^{-1}$) for only 5 min followed by 175 min of exposure to medium without PG, or exposure to PG for 180 min induced similar levels of 3D biofilm formation. In all panels (**a**–**g**), bars are mean values of $n = 3$ independent biological replicates, circles indicate individual measurements and error bars indicate the standard deviation. Statistical significances were calculated as indicated by the black lines (in **a**, **e** and **f**) or relative to the lysate condition (in **b**) using a one-way ANOVA with Bonferroni's correction; in panels **c** and **g**, a two-sided Student's *t*-test was used. Statistical results are given as exact *P* values in brackets in the graphs or indicated using the following: NS, not significant ($P > 0.99$, unless specified otherwise in the figure), ****$P < 0.0001$.

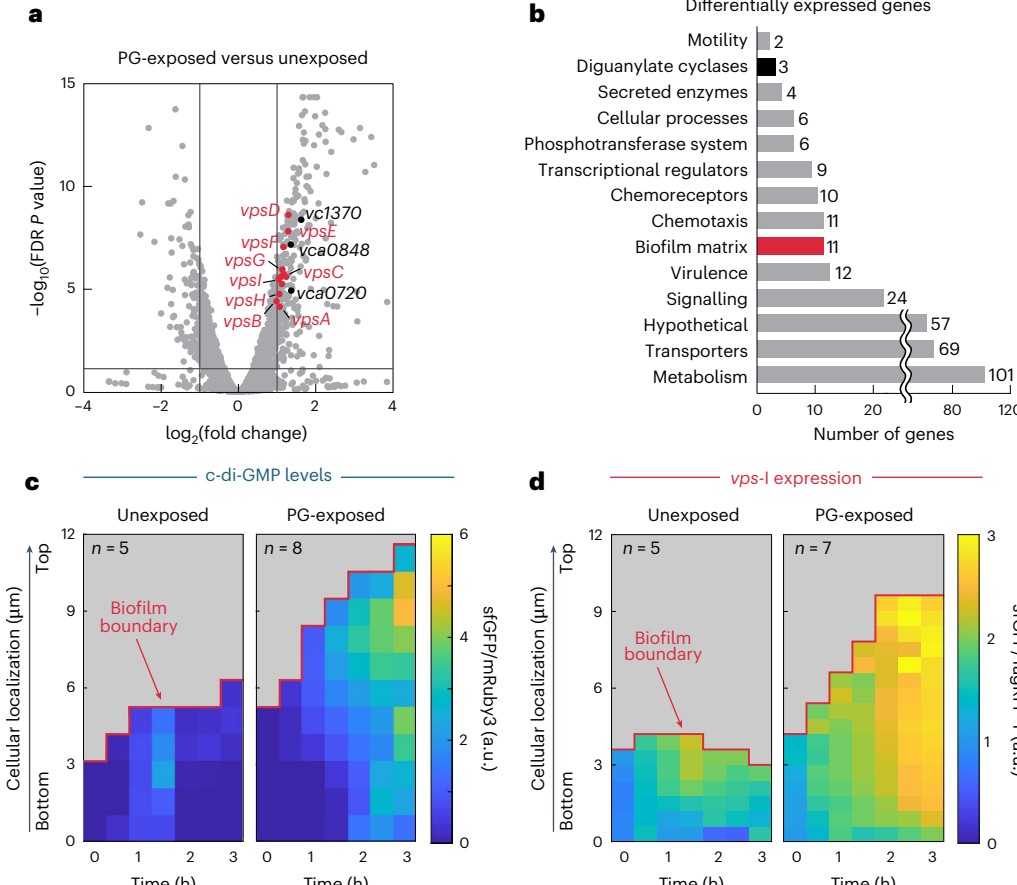

**Fig. 4 | The transcriptional response to exogenous PG increases c-di-GMP levels and biofilm matrix production. a**, Transcriptome comparison of *V. cholerae* WT cells that were exposed to PG for 10 min (300 µg ml⁻¹ in LB) or to the control condition (LB medium without PG) for the same time (*n* = 3 for each condition). Genes with absolute fold changes >2 and a FDR-adjusted *P* < 0.05 were considered to be differentially expressed (see Source Data Fig. 4 for a complete list). Genes were functionally categorized using annotations from UniProt[62], Kyoto Encyclopedia of Genes and Genomes[63] and MicrobesOnline[64]. **b**, Quantification of the number of genes upregulated during 10 min of PG exposure, for different functional categories, using the same colour scheme as in **a**. **c**, Spatiotemporal measurements of a fluorescent c-di-GMP reporter in *V. cholerae* WT cells treated with PG-exposure (300 µg ml⁻¹, blue) or the control

condition (LB without PG, black) over 3 h. The c-di-GMP reporter was quantified as the fold change in unstable sfGFP fluorescent intensity levels normalized by the fluorescence intensity of a constitutive reporter (P$_{tac}$-*mRuby3*). High levels of sfGFP indicate high c-di-GMP levels (see calibration in Extended Data Fig. 9a). The spatiotemporal heat maps indicate averages of *n* = 5 (unexposed) and *n* = 8 (PG-exposed) biological replicates. **d**, Spatiotemporal measurements of a sfGFP-based fluorescence reporter for *vps*-I operon transcription in *V. cholerae* WT. Measurements were performed over 3 h in the presence or absence of pure PG (300 µg ml⁻¹). The fluorescence of the sfGFP-based transcriptional reporter was normalized by the fluorescence of a constitutive reporter (P$_{tac}$-*TagRFP-T*). The spatiotemporal heat maps indicate averages of *n* = 5 (unexposed) and *n* = 7 (PG-exposed) biological replicates.

Among the nine upregulated transcriptional regulators are HapR and RpoS, which are both linked to the regulation of matrix production[32], and TfoX, which is linked to competence and DNA uptake[33]. It is worth noting that three genes coding for diguanylate cyclases (genes *vc1370*, *vca0720/hnoX*, *vca0848*) were upregulated following PG exposure (Fig. 4b). Diguanylate cyclases can increase the intracellular levels of cyclic dimeric guanosine monophosphate (c-di-GMP) which can further increase the production of extracellular matrix components beyond VPS[32,34], and a previous study has shown that ectopic expression of *vc1370* or *vca0848* causes biofilm formation[35].

We used fluorescence-based reporters and live-cell confocal microscopy to test whether we could confirm the key results from the transcriptome analysis (Fig. 4c,d). To measure the level of c-di-GMP, we adapted a reporter based on three consecutive riboswitches (Bc3–Bc5 from *Bacillus thuringiensis*[36,37]) that regulate the expression of an unstable superfolder-GFP (including the LAA degradation tag) in a c-di-GMP-dependent manner, harboured on a low-copy number plasmid. When c-di-GMP is bound to the Bc3–Bc5 riboswitches, the messenger RNA coding for the unstable sfGFP can be translated,

resulting in fluorescence (validation, Extended Data Fig. 9a). Using this reporter, we observed that c-di-GMP levels substantially increase in our flow chamber system following exposure to exogenous PG (Fig. 4c). Increased levels of c-di-GMP generally induce the expression of extracellular matrix genes, causing biofilm formation[32,34,38]. To measure the spatiotemporal expression of the *vps*-I operon in the presence or absence of exogenous PG, we generated a fluorescent transcriptional reporter based on the insertion of *sfgfp* into the native *vps*-I operon. These measurements revealed that *vps*-I transcription was increased when the cells were exposed to exogenous PG (Fig. 4d). Despite our identification of the upregulation of VPS production and increase in c-di-GMP levels, which cause 3D biofilm formation, it remains unclear how *V. cholerae* senses exogenous PG.

## PG-induced biofilms protect against phages
To test whether the 3D biofilms formed in response to exogenous PG protect the cells from phage infection, we grew *V. cholerae* in microfluidic channels for 1 h with or without exogenous PG, followed by the addition of purified Vibriophage N4. To measure the extent of phage

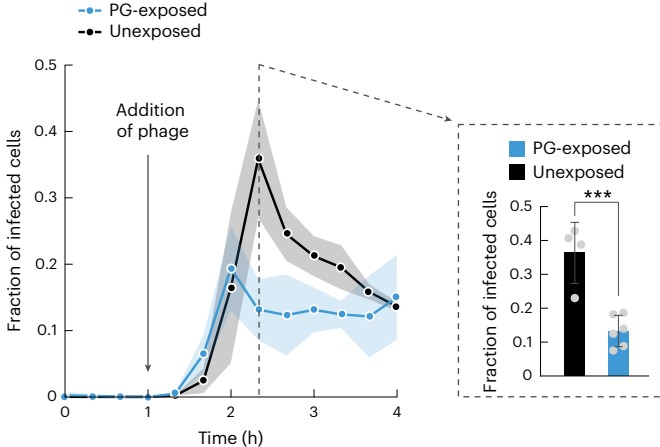

**Fig. 5 | Biofilm formation in response to exogenous PG protects against phage predation.** Using a phage infection reporter and microscopy, the fraction of infected cells was quantified. PG-exposed cells were less susceptible to infection by $10^6$ p.f.u. ml$^{-1}$ Vibriophage N4 than unexposed cells. PG exposure was initiated at time $t = 0$ h, phage exposure was initiated at $t = 1$ h. Lines indicate the mean of biological replicates ($n = 6$ for PG-exposed, $n = 4$ for unexposed), and shaded regions represent the standard deviation. Bar graph shows the difference between PG-exposed and unexposed cells at the time of peak phage infection. Statistical significance was calculated using a two-sided Student's $t$-test; ***$P = 0.0007$.

infection in this system, we constructed a fluorescent protein infection reporter based on the promoter of the major phage capsid protein VN4_32 (Extended Data Fig. 9b–d). These experiments showed that the cells inside biofilms, which formed as a consequence of PG exposure, are protected from phage infection (Fig. 5). Control experiments, in which 3D biofilm formation was achieved independent of PG exposure (by genetically inducing elevated c-di-GMP levels), showed that 3D biofilm formation generally protects against phage infection (Extended Data Fig. 10). Biofilms are therefore a niche where phage-susceptible bacteria can survive without the need for phage receptor mutations.

### PG is a danger signal across different species

As exogenous PG is not toxic for *V. cholerae* (Extended Data Fig. 5b), the biofilm formation response cannot be interpreted as a protective response against exogenous PG. For other species, exogenous PG can even be growth promoting[39,40]. The response of *V. cholerae* to exogenous PG is therefore qualitatively distinct from biofilm induction by cell damage-inducing compounds, such as antibiotics[41,42]. Exogenous PG can indicate the presence of ecological competition[43], and for *P. aeruginosa* exogenous PG indeed induces a competition response in the form of pyocyanin secretion[44]. However, phage predation and abiotic stresses can also lead to the release of PG so that PG is not only a competition signal but more generally a danger signal. As there are many ways in which cells can lyse, and we found that lysate from several different species triggered *V. cholerae* biofilm formation (Fig. 3a), we conclude that exogenous PG is a general danger signal for *V. cholerae*.

Based on the results for *V. cholerae*, we tested whether exogenous PG could also be a danger signal for other species. We explored the response of several bacterial species to purified exogenous PG, using the same flow chamber-based cultivation system that was used for *V. cholerae* in Figs. 1–4. Our flow chamber system requires that bacterial species attach to the glass surface of the channel before stimulation with exogenous PG. As some species do not attach to glass in our particular conditions, it was not possible to test a representative set of species from the bacterial phylogenetic tree. However, several Gram-negative and distantly related Gram-positive pathogens could be assessed: *P. aeruginosa*, *Acinetobacter baumannii*, *E. coli*,

*S. aureus*, *Staphylococcus epidermidis* and *Enterococcus faecalis*. While the responses of *E. coli* and *S. epidermidis* were not statistically significant, all other species responded to exogenous PG by forming 3D biofilms (Fig. 6). These results show that exogenous PG is a widely used danger signal among bacteria, which triggers biofilm formation as a protective measure against various forms of biotic or abiotic stresses.

## Discussion

Analogous to our finding that bacteria use exogenous PG as a danger signal, human immune cells use PG subunits as danger signals. These processes have been characterized in detail for the pattern recognition receptors NOD1 and NOD2 (ref. 45), which detect meso-diaminopimelic acid containing PG and the muramyl dipeptide, respectively. In fact, the repertoire of danger signals that have been identified for human immune cells is currently much larger than the set of danger signals that are known for bacteria[45–47]. For *P. aeruginosa*, it has been observed that specifically the lysis of kin cells triggers the activation of type 6 secretion system via the Gac/Rsm signal transduction pathway[2,48], yet the precise signal molecule involved in this process remains unknown. During the revision of our manuscript, it was reported that *V. cholerae* can respond to norspermidine released by kin cells and closely related Vibrios[49]. For *P. aeruginosa* swarms, it has also been observed that phage infection and antibiotic treatment trigger the release of the *Pseudomonas* quinolone signal molecule which then repels other conspecific *P. aeruginosa* cells in the swarm that have not been exposed to these dangers[3]. Analogously, phage infection of *B. subtilis* elicits a phage tolerance in neighbouring kin cells[4]. In *E. coli* swarms, antibiotic-induced cell lysis can release a part of a resistance-nodulation-division efflux pump that can then be used by kin cells to stimulate antibiotic efflux of existing pumps, and it induces the production of additional efflux pumps[50]. In contrast to these other bacterial danger-sensing systems, we found that exogenous PG is a danger signal very broadly used not only between kin cells but also between widely different species (Fig. 6). Consistent with exogenous PG being a general danger signal, the response we observed to exogenous PG, that is biofilm formation, results in a general protection against many threats to bacterial survival.

In this study, we have demonstrated that exogenous PG released by cellular lysis is a general danger signal to which several bacterial species respond by forming biofilms. The resulting biofilms serves as a refuge that protects against phage predation or other biotic and abiotic stresses that can lyse bacterial cells. Sensing exogenous PG provides cells that are in the vicinity of phages, or in the vicinity of other stresses that lead to lysis, a strategy for sensing imminent danger before direct contact with the stress. Only mechanisms that kill bacteria without lysis or by phagocytosis would prevent the release of this danger signal. The large number of genes that were upregulated or downregulated in *V. cholerae* in response to brief exposure to exogenous PG, including major regulatory and virulence genes, as well as phage defence genes, suggests that this danger signal could result in several additional protective phenotypes that go beyond biofilm formation. Understanding the full scope of the responses to exogenous PG promises to reveal basic new insights into bacterial stress responses and danger sensing.

## Methods

### Bacterial strains and culture conditions

All *V. cholerae* strains used in this study are derivatives of the WT O1 biovar El Tor strain C6706. *V. cholerae*, *E. coli*, *P. aeruginosa*, *B. subtilis*, *E. faecalis* and *A. baumannii* strains were cultured in LB Miller medium (Roth) at 37 °C with shaking at 250 r.p.m. (revolutions per minute). *S. epidermidis* and *S. aureus* strains were cultured in tryptic soy broth (TSB) at 37 °C with shaking at 250 r.p.m. Tryptone broth contained 10 g l$^{-1}$ tryptone and 10 g l$^{-1}$ NaCl. M9 medium contained M9 minimal salts (M6030, Sigma) supplemented with 2 mM MgSO$_4$, 100 mM CaCl$_2$, minimum essential medium (MEM) vitamins, 15 mM triethanolamine (pH 7.1), 0.6 μM FeCl$_3$ and 0.5% glucose. Where required, the culture

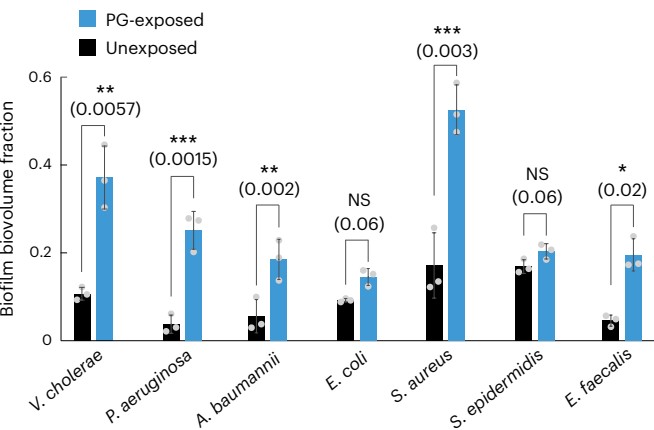

**Fig. 6 | Exogenous peptidoglycan is a conserved signal for inducing biofilm formation in different species.** Biofilm formation of different species was quantified after 3 h of exposure to PG (300 µg ml⁻¹, blue bars) or control conditions (medium without PG, black bars). Among the species are *V. cholerae* (as a control, similar to Fig. 3e–g), as well as other Gram-negative pathogens (*P. aeruginosa*, *A. baumannii*, *E. coli*) and Gram-positive pathogens (*S. aureus*, *S. epidermidis*, *E. faecalis*). For *E. coli* and *S. epidermidis*, the presence or absence of PG did not result in statistically significant differences in 3D biofilm formation. All other species showed enhanced 3D biofilm formation during PG exposure. Bars are mean values of $n = 3$ independent biological replicates, circles indicate individual measurements, and error bars indicate the standard deviation. Statistical significances were calculated using a two-sided Student's *t*-test. Statistical results are given as exact *P* values in brackets in the graphs.

medium was supplemented with the following antibiotics: gentamicin (30 µg ml⁻¹), ampicillin (100 µg ml⁻¹ for *E. coli* and 200 µg ml⁻¹ for *V. cholerae*) or kanamycin (50 µg ml⁻¹ for *E. coli* and 100 µg ml⁻¹ for *V. cholerae*). Detailed lists of strains and plasmids are provided in Supplementary Tables 1 and 2, respectively.

## Plasmid construction

Plasmid construction was carried out using standard molecular biology techniques[51]. All enzymes for cloning were purchased from New England Biolabs or Takara Bio. Primers were designed using the SnapGene software v.4.2.11 (Insightful Science). Oligos used for plasmid construction are listed in Supplementary Table 3 and were synthesized by Eurofins or Sigma Aldrich. All constructed plasmids were verified by Sanger sequencing, which was carried out by Eurofins or Microsynth SeqLab.

## Gene deletions in *V. cholerae*

To delete a chromosomal gene in *V. cholerae*, a plasmid based on the suicide vector pKAS32 was generated for the respective gene[52]. Briefly, the vector pNUT144 (a derivative of pKAS32) was amplified using oligos KDO1968 and KDO1969. Upstream and downstream, flanking regions of the gene of interest (1 kb each) were amplified with suitable oligos (listed in Supplementary Table 3) using genomic DNA of strain KDV201. The final plasmid, comprising the amplified vector backbone and inserts, was constructed by Gibson assembly. The final plasmid was harboured in *E. coli* S17-λ pir. This method was used to generate the plasmid pNUT2259, which was introduced into KDV428 by conjugation to create the *V. cholerae* ΔtrxA mutant (strain KDV2489). A similar protocol was used to generate the strains with ΔcdgDΔcdgKΔcdgHΔcdgL or ΔrocSΔcdgJ mutations (strains KDV2849 and KDV2824).

## Fluorescent reporter strains

For the phage infection reporter, the fragment P_{VN4_32}-*mNeonGreen* was constructed. To this end, the putative promoter of the gene encoding the major capsid protein (*VN4_32*) was amplified from Vibriophage N4 DNA (directly from a phage lysate), using oligos KDO1519 and KDO1520. The gene encoding the fluorescent protein mNeonGreen was amplified

from the plasmid pNUT1035 using primers KDO1521 and KDO1522. The two fragments were then fused together by overlap polymerase chain reaction using primers KDO1534 and KDO1535. The resulting P_{VN4_32}-*mNeonGreen* fragment was digested with restriction enzymes NheI and PacI (New England Biolabs) and ligated into a similarly digested vector pNUT480 (which is a derivative of pKAS32). The resulting plasmid pNUT1532 contained the P_{VN4_32}-*mNeonGreen* fragment, flanked by 1 kb regions that are homologous to the up-stream and down-stream sequences of *lacZ* on the chromosome. Plasmid pNUT1532 was then introduced into KDV201 by conjugation, yielding strain KDV986, in which *lacZ* was replaced by the P_{VN4_32}-*mNeonGreen* construct. For the constitutive expression of a fluorescent protein in the phage infection reporter strain, the plasmid pNUT1475 was constructed. This plasmid has a pSC101* vector background and a gentamicin resistance cassette and includes the gene encoding the red fluorescent protein TagRFP-T under the control of a P_{tac} promoter. To generate this plasmid, the *TagRFP-T* fragment was amplified from the plasmid pNUT1442 using primers KDO1599 and KDO719, digested with BamHI and SacI (New England Biolabs) and subsequently ligated with the backbone of a similarly digested plasmid pNUT1027 to create the plasmid pNUT1475. This plasmid was harboured in an *E. coli* Top10 strain and introduced into KDV986 by triparental mating (including an *E. coli* helper strain harbouring the plasmid pRK600) to create the final strain KDV992.

To generate the c-di-GMP reporter plasmid pNUT3038, which is a riboswitch-based fluorescent protein reporter for intracellular c-di-GMP levels[36], the DNA sequences encoding the naturally occurring riboswitches Bc3–Bc5 were amplified from *B. thuringiensis* subsp. *chinensis* CT-43 using oligos KDO3443 and KDO3769. The riboswitches were cloned upstream of *sfGFP*, which was amplified from pNUT480 using oligos KDO3445 and KDO3446 and fused with *bc3-5* using overlap polymerase chain reaction. To add a constitutive fluorescent protein expression construct to the plasmid, P_{tac}-*mRuby3* was amplified from pNUT1029 using oligos KDO3442 and KDO3552. The two amplified fragments were cloned into a low-copy-number plasmid with a pSC101* origin of replication and a gentamicin resistance cassette, whose backbone was amplified using oligos KDO3441 and KDO3828 from pNUT1029. The resulting plasmid pNUT2828 was constructed using Gibson assembly. To visualize dynamic changes in c-di-GMP, a degradation tag with amino acid sequence LAA[53] was added to the sfGFP fluorescent protein. For this, the primer KDO3688 contained an overhang sequence encoding the degradation tag, which was used along with KDO3689 to amplify the P_{tac}-*mRuby3_bc3-5-sfGFP* fragment from pNUT2828. The plasmid backbone of pNUT2828 was amplified with oligos KDO3690 and KDO3691. The two fragments were fused by Gibson assembly to create the final plasmid pNUT3038. This plasmid was harboured in an *E. coli* Top10 strain and introduced into KDV201 by triparental mating (including an *E. coli* helper strain harbouring the plasmid pRK600) to create the final strain KDV2971.

## Flow chamber experiments

Bacterial cells were grown in LB medium (*V. cholerae*, strains KDV428, KDV504, KDV992, KDV2151, KDV2489, KDV2971; *P. aeruginosa*, strain KDP39; *E. coli*, strain KDE1469, *A. baumannii* KDM131, *E. faecalis* KDM129) or TSB (*S. epidermidis*, strain KDM115; *S. aureus*, strain KDM2) overnight at 37 °C with shaking at 250 r.p.m. Day cultures were prepared by diluting the overnight cultures 1:200 in fresh medium and incubating the cultures at 37 °C under shaking conditions until optical density at 600 nm (OD₆₀₀) = 0.4. These bacterial cells were inoculated into microfluidic flow chambers made from polydimethylsiloxane (PDMS) and glass coverslips, as described in detail previously[54,55]. The specific protocol we used is as follows: PDMS and glass coverslips were bonded using an oxygen plasma, resulting in flow chambers that were 7,000 µm in length, 500 µm in width and 100 µm in height. The microfluidic design contained either four or eight independent channels of identical dimensions, each with its own inlet and outlet, on

a single chip. The manufacturing process of these microfluidic channels guarantees highly reproducible channel dimensions and surface properties in the channels.

Following inoculation of the channels, the cells were given 1 h to attach to the glass surface of the channel without flow. For flow chamber experiments with *V. cholerae* cells, the microfluidic chambers were then connected via polytetrafluorethylene tubing to syringes containing LB supplemented with either (1) purified Vibriophage N4, (2) bacterial lysates, (3) isolated DNA from *V. cholerae* cells or (4) pure PG. For flow chamber experiments with *E. coli*, *P. aeruginosa*, *A. baumannii* and *E. faecalis*, the microfluidic chambers were connected to syringes containing LB supplemented with pure PG. For flow chamber experiments with *S. epidermidis* or *S. aureus*, the microfluidic chambers were connected to syringes containing TSB supplemented with pure PG. A flow of 100 μl min⁻¹ through each microfluidic channel was then initiated for 45 s to wash away non-adherent cells. The flow rate was then set to 0.1 μl min⁻¹ until the end of the experiment, and the flow chambers were incubated in a 37 °C incubator. Flow rates were controlled using a high-precision syringe pump (Harvard Apparatus). The channels were imaged on an inverted confocal fluorescence microscope, through the coverslip at the bottom of the microfluidic channels. For all bacterial species, confocal imaging relied on constitutively expressed fluorescent proteins, except for *S. aureus* (strain KDM2) and *A. baumannii* (strain KDM131), which was visualized by staining with 4 μM SYTO9 (Thermo Fisher, S34854) after biofilm growth.

## Phage amplification, purification and inactivation

To propagate the *V. cholerae* bacteriophage N4 (strain ATCC 51352-B1), *V. cholerae* C6706 WT (KDV201) was used as the bacterial host. Phage lysates were prepared using a protocol described previously[14]. The specific protocol we used is as follows: Cultures of KDV201 were grown in LB at 37 °C with shaking until $OD_{600} = 0.4$ and infected with phages (from a frozen phage lysate) at a MOI of 0.1. Bacteria and phages were incubated together for 1 h or until the culture became clear, as a result of bacterial lysis following phage infection. The lysate was filtered using a 0.22 μm filter (Roth) and stored at 4 °C (for short-term storage) or −80 °C (for long-term storage). The phage titre of the lysates was ~10⁹ p.f.u. ml⁻¹ (as determined by a plaque assay).

For infection experiments, purified phages were used. The phage particles were purified using a previously described method[56], with minor modifications. The protocol is as follows: Phage lysate was prepared by growing cultures in brain heart infusion broth until $OD_{600} = 1.2$, followed by the infection of this culture with previously prepared phage lysate at an MOI of 0.1. The resulting lysate was treated with DNase I (1 μg ml⁻¹) for 30 min at 37 °C with shaking. NaCl was added (0.5 g ml⁻¹), and the treated lysate was stored at 4 °C for 1 h. The lysate was then filtered using a 0.22 μm filter (Roth), and phages were precipitated using PEG 6000 (10% *w/v*) for 2 days at 4 °C. The precipitated phages were collected by centrifugation at 7,800 × *g* for 15 min at 4 °C. Phages were then purified using a CsCl density gradient column. The column was prepared using the CsCl:PBS ratios described for T7 *E. coli* phage in the product manual of the T7 Select Novagen kit and centrifuged at 100,000 × *g* at 4 °C for 24 h. No visible band of concentrated phages was observed, but the layer in which the T7 phages would have been expected was acquired and used for further purification. The CsCl was removed by dialysis with PBS using a 14,000 Da membrane filter. The titre of the purified phages was 10¹⁰ p.f.u. ml⁻¹. Purified phages were stored at 4 °C.

To inactivate purified phages, a heat treatment was applied[24]. For this, purified phages in PBS were heated to 65 °C for 15 min, then cooled down to 37 °C before incubation with *V. cholerae* at 37 °C.

## Quantifying phage susceptibility of biofilm populations

To test whether continuous phage exposure for 8 h in flow chambers resulted in a substantial fraction of phage-resistant cells in the biofilm population, strain KDV201 was exposed to Vibriophage N4 in flow

chambers for 8 h. Bacterial cells were then collected from the flow chambers by mechanical scraping from the glass bottom surface and the PDMS top surface using a razor blade, followed by resuspension in fresh LB using vigorous vortexing and two washes with fresh LB medium. Despite the washes, some phages remained in the suspension or remained adsorbed to cells or remained inside cells. Cells from this collected population were co-incubated with or without fresh phages to test for their susceptibility to phage infection. Changes in bacterial culture density were monitored using a microplate reader (Spark 10 M, Tecan) under shaking conditions at 37 °C.

## Crystal violet assay

To test the biofilm formation capability of bacteria that were exposed to phages for 8 h in flow chambers, a crystal violet assay was performed using a protocol described previously[57]. The specific protocol we used is as follows: Strain KDV201 was exposed to Vibriophage N4 in flow chambers for 8 h, at which point the cells were collected as described above, which resulted in a cell suspension that retained some phages. The collected cell population was streaked out on LB agar plates and incubated at 37 °C for 24 h. On the following day, isolated colonies were inoculated in 180 μl of fresh LB (in a 96-well plate). As controls, the *V. cholerae* WT (KDV201), Δ*vpsL* (KDV207) and *vpvC*^W240R (rugose; KDV941) strains were also inoculated into individual wells of a 96-well plate from colonies on LB agar. These strains served as two negative controls and a positive control, respectively. For each strain, bacteria from a single colony were inoculated in triplicates into a 96-well plate. The microtitre plate was incubated at 37 °C with shaking (810 r.p.m.) inside a microplate reader until $OD_{600} = 0.4$. These cultures were then diluted 1:2,000 in 150 μl of fresh LB into another 96-well plate, which was incubated for biofilm growth at 25 °C for 14 h. After this incubation period, the culture was discarded and the wells washed to remove any unattached bacterial biomass. A 0.1% solution of crystal violet in water was used to stain the surface-attached bacterial biomass. After 15 min of incubation at room temperature, the crystal violet solution was discarded, and the wells were washed twice in clean water to remove any excess dye. The plates were left to dry upside down overnight at room temperature. On the following day, 30% acetic acid was added to each well to solubilize the crystal violet, and the optical density was measured at a wavelength of 550 nm ($OD_{550}$) using a microplate reader. For each measurement, the data were averaged from three wells (technical replicates) per experiment, and *n* = 3 independent biological replicates were performed.

## Colony rugosity assay

To test whether continuous phage exposure for 8 h in flow chambers resulted in a high frequency of mutants that produce a high amount of extracellular matrix (termed 'matrix hyper-producers' or strains with a 'rugose' colony morphology), cells were collected from the flow chambers after 8 h of phage exposure as described in the Methods section 'Crystal violet assay'. The collected cells were streaked out on LB agar plates and incubated at 37 °C for 24 h. The following day, isolated colonies were inoculated in 200 μl of fresh LB and spotted on LB agar plates, which were incubated at room temperature for 3 days. To retain humidity, the plates were sealed with parafilm. As controls, the *V. cholerae* WT (KDV201), Δ*vpsL* (KDV207) and *vpvC*^W240R (rugose strain, KDV941) overnight cultures were also spotted on LB agar plates as two negative controls and a positive control, respectively. High matrix production (also known as rugosity) in *V. cholerae* manifests in the form of wrinkled and rough-looking bacterial colonies on LB agar[58]. Morphologies of colonies grown from the collected cells were visually compared with the smooth colonies of the WT and Δ*vpsL* strain and the wrinkled colonies of the rugose *vpvC*^W240R strain.

## Phage adsorption assay

To measure the Vibriophage N4 adsorption to *V. cholerae* WT or Δ*trxA* cells, the unabsorbed phages in the supernatant were enumerated by

performing plaque assays at different times during phage infection. Cultures of strains KDV201 or KDV2464 were grown in LB at 37 °C until $OD_{600} = 0.4$ and exposed to purified phages at a MOI of 0.001. Bacteria and phages were co-incubated at 37 °C with shaking (250 r.p.m.). From this culture, 450 µl was sampled at 0, 4, 8 and 16 min after phages were added. Each sample was immediately centrifuged at $10,000 \times g$ for 2 min. The supernatants were transferred to fresh tubes, which were placed on ice until all samples were collected. Phage-containing supernatants were serially diluted (up to $10^{-3}$) in PBS and their phage titres enumerated by a plaque assay. A decrease in the concentration of phages in the supernatant within the first few minutes after phage exposure indicates adsorption of phages to bacteria. For each experiment, plaque-forming unit measurements were averaged from three LB agar plates per bacterial strain (technical replicates), and a total of $n = 3$ biological replicates were performed.

### EOP assay
To quantify the susceptibility of *V. cholerae* WT (strain KDV201) or Δ*trxA* (strain KDV2489) cells to Vibriophage N4 infection, a relative EOP assay was performed. Purified phages were serially diluted from $10^8$ p.f.u. ml$^{-1}$ to $10^4$ p.f.u. ml$^{-1}$ in LB. Of each phage dilution, 10 µl was spotted on bacterial lawns of strains KDV201 or KDV2489 that were grown on an LB agar plate. The plates were incubated at 37 °C overnight. On the following day, the number of plaque-forming units per millilitre were enumerated for each spotted phage inoculum. The EOP was calculated as the ratio of the plaque count and the number of virions in the given spotted phage inoculum. For each experiment, measurements were averaged from 5 phage spots per bacterial strain and phage dilution (technical replicates), and a total of $n = 3$ biological replicates were performed.

### Preparation of sonicated bacterial lysates
Bacterial lysates were prepared from several Gram-negative species (*V. cholerae* strain KDV201, *P. aeruginosa* strain KDP43, *E. coli* strain KDE474) and Gram-positive species (*B. subtilis* strain KDB2, *S. aureus* strain KDM2). Bacterial cultures of all species were grown in LB medium, except for *S. aureus*, which was grown in TSB. The cultures were incubated at 37 °C with shaking at 250 r.p.m. Bacterial overnight cultures were diluted 1:100 in their respective fresh growth medium and grown at 37 °C with shaking until $OD_{600} = 0.4$. These cells were washed twice with equal volumes of fresh medium and then concentrated 100 times into fresh medium. Sonication was performed on ice using an ultrasonic probe (Heilscher UP200St) with settings of 50% capacity (0.5 s on and 0.5 s off), 80% amplitude, 1 min intervals with 1 min rest for 45 min. Cells were enumerated by plating on LB agar before and after sonication to determine the fraction of cells that were lysed. Lysates prepared with this protocol yielded ~99% lysis for all species. After sonication, the raw lysate was centrifuged ($9,000 \times g$ for 10 min at 4 °C) and filtered through a 0.22 µm filter (Roth) to remove intact bacterial cells. Bacterial lysates were stored at −80 °C.

To narrow down which part of the lysate contained the biofilm-inducing factor, sonicated lysates made from *V. cholerae* cells were treated one or more of the following enzymes: DNase I (Thermo Fisher, 18047019) at a final concentration of 1 U ml$^{-1}$ incubated at 37 °C for 30 min, RNase A (Thermo Fisher, EN0531) at a final concentration of 1 µg ml$^{-1}$ incubated at 37 °C for 30 min and proteinase K (Roth, 7528.1) at a final concentration of 20 µg ml$^{-1}$ incubated at 37 °C for 60 min. Lysates treated with these enzymes were then flowed through microfluidic chambers that were inoculated with *V. cholerae* strain KDV428.

To roughly characterize the molecular weight of the biofilm-inducing factor, sonicated lysates made from *V. cholerae* cells were filtered through membrane filters of varying pore sizes (3 kDa, 10 kDa, 30 kDa, 100 kDa and 300 kDa, all from Merck Millipore). The filtrates were then flown into microfluidic chambers that were previously inoculated with *V. cholerae* strain KDV428.

### Preparation of spheroplast lysate
Spheroplasts were prepared using a protocol described previously[29]. The specific protocol we used is as follows: Exponentially growing cells of *V. cholerae* C6706 (KDV201) grown in LB at 37 °C until $OD_{600} = 0.4$ were collected by centrifugation at $5,000 \times g$ for 5 min. The cell pellet was washed with an equal volume of LB once, followed by two washes with 10 mM Tris–HCl (pH 8.0). Cells were then resuspended in spheroplasting buffer, using 1/5 of the initial volume (resulting in a 5× concentration). This buffer solution is composed of 10 mM Tris–HCl (pH 8.0), 10 mM EDTA and 20% (*w/v*) sucrose. Lysozyme (Sigma, 1052810010) was added to achieve a final concentration of 1 mg µl$^{-1}$, and the cells were incubated at room temperature with gentle shaking for 3 h to generate spheroplasts, which was confirmed using microscopy. Spheroplasts were pelleted down by centrifugation at $16,000 \times g$ for 10 min and resuspended in LB, using 1/100 of the initial volume (resulting in a 100× concentration). This suspension was immediately sonicated on ice to obtain spheroplast lysate, using the sonication settings described above.

### Assays with PG
Purified *B. subtilis* PG was purchased from Sigma (69554-10MG-F) and suspended in fresh LB at a final concentration of 300 µg ml$^{-1}$. Soluble PG was obtained by sonicating insoluble commercial PG, using the sonication settings described above for obtaining bacterial lysate. After sonication, undissolved PG was pelleted by centrifugation, and the supernatant was used for the experiments.

To purify the cell wall from *V. cholerae* C6706 (KDV201) for Fig. 3c, the cell wall was isolated using a previously described protocol[59]. The specific protocol we used is as follows: *V. cholerae* cells from an overnight culture ($OD_{600} = 4.0$) were collected by centrifugation at $5,000 \times g$ for 5 min. The cell pellet was washed with equal volume of LB once and then resuspended in PBS, using 1/10 of the initial volume. Resuspended cells were added drop-wise into 10 ml of boiling 10% sodium dodecyl sulfate (SDS) (in a water bath), while continuously stirring. This suspension was boiled for 2–3 h (water was replenished to prevent drying). Cell wall material was pelleted by ultracentrifugation ($2.7 \times 10^5 \times g$ for 10 min at 20 °C). The pellet was washed with double-distilled $H_2O$ three times. Finally, the pellet was resuspended in LB using 1/10 of the initial volume. This suspension was immediately sonicated using the sonication settings described above to obtain a crude extract of PG fragments, which was used for the experiments only in Fig. 3c.

### Protein expression and purification
The *P. aeruginosa* and *V. cholerae* genes encoding AmpDh3$_{PA14}$, ShyA$_{Vc}$ and MltA$_{Vc}$ were cloned on pET28b(+) (Novagen) with C-terminal His-tags for expression in *E. coli* BL21 (DE3) cells. Bacteria were cultured in Terrific Broth (24 g l$^{-1}$ yeast extract, 20 g l$^{-1}$ tryptone, 4 ml l$^{-1}$ glycerol, 0.017 M KH$_2$PO$_4$, 0.072 M K$_2$HPO$_4$), and expression was induced at $OD_{600}$ 0.4 with 1 mM isopropyl-β-D-thiogalactopyranoside and left overnight at 16 °C. Cell pellets were resuspended in PBS with a Complete Protease Inhibitor Cocktail Tablet (Roche) and lysed by two passes through a French press at 68.9 MPa (10,000 p.s.i.) After centrifugation (30 min, $100,000 \times g$), proteins were purified from the cleared lysates via Ni-NTA agarose columns (Qiagen) and eluted with a discontinuous imidazole gradient using an ÄktaGo system. Purified fractions were loaded on a size exclusion chromatography Superdex 200 Increase 10/300 GL column equilibrated with 100 mM citrate/citric acid buffer (pH 5) with 300 mM NaCl. Purified proteins were visualized by SDS–PAGE electrophoretic protein separation and quantified by Bio-Rad Protein Assay (Bio-Rad). The proteins were either stored at 4 °C for immediate use or at −80 °C after the addition of 10% (*v/v*) glycerol.

### In vitro production of PG digests
The lysozyme, amidase (AmpDh3$_{PA14}$), D,D-endopeptidase (ShyA$_{Vc}$) and lytic transglycosylase (MltA$_{Vc}$) reactions were performed in 3 ml

reactions using 50 µg ml⁻¹ of purified enzymes with 300 µg ml⁻¹ of sacculi isolated as described above from stationary phase cultures from *V. cholerae* (C6706). All digestions were carried out in 50 mM sodium phosphate buffer (pH 4.9) overnight at 37 °C. Reactions were heat-inactivated (boiled for 5 min), and fractions were separated by centrifugation at 20,000 × *g* for 15 min. The soluble fraction was used for the experiments, 10 µl of which was subjected to sample reduction to check correct digestion by liquid chromatography–mass spectrometry. First, pH was adjusted to 8.5–9 by the addition of borate buffer (0.5 M, pH 9) and then *N*-acetylmuramic acid residues were reduced to muramitol by sodium borohydride treatment (NaBH₄ 10 mg ml⁻¹ final concentration) for 30 min at room temperature. Finally, pH was adjusted to 2.0–4.0 with orthophosphoric acid 25% (*v*/*v*) before analysis by liquid chromatography–mass spectrometry.

### Sample collection for RNA-seq
To collect a sufficient amount of biomass for RNA-seq, we inoculated *V. cholerae* C6706 (KDV201) cells in flow channels of six separate identical microfluidic chips, using the procedure described above. After exposure to PG (300 µg ml⁻¹, sonicated) dissolved in LB or the control condition (LB without PG) for 10 min, the microfluidic device was disassembled, and a 1:1 mixture of PBS and RNAstop solution (a mixture of 95% (vol/vol) EtOH and 5% (vol/vol) phenol) was flowed across the glass surface to terminate transcription and translation. Subsequently, bacterial biomass was collected by scraping cells off of the glass surface using a clean razor blade, to obtain 1–2 ml of cell suspension, which was added to a 2 ml Eppendorf tube, and excess RNAstop solution (supernatant) was removed after centrifugation at 4 °C. Cells from each sample were resuspended in 50 µl of lysozyme buffer, which consisted of TE buffer (composed of 10 mM Tris (adjusted to pH 8.0 with HCl) and 1 mM ethylenediaminetetraacetic acid (EDTA)) and 20 U µl⁻¹ Ready-lyse lysozyme (Lucigen, R1804M). These suspensions were immediately snap-frozen in liquid nitrogen and stored at −80 °C until RNA isolation was performed. This sample collection procedure was performed on three separate days, to obtain a *n* = 3 biological replicates for each of the two conditions.

### RNA isolation and sequencing
Total RNA was extracted using the hot SDS/hot phenol method[60] with the following modifications. Cells were lysed at 65 °C for 2 min in the presence of 1% (*w*/*v*) SDS. Then, 6 µl of 1 M sodium acetate (pH 5.5) and 62.5 µl of Roti-Aqua-Phenol (Roth, A980) were added to the lysate and incubated at 65 °C for 8 min. The whole mixture was transferred to a phase lock gel tube (VWR, 733-2478), followed by the addition of 62.5 µl of chloroform (Sigma, C2432). The mixture was centrifuged at 20,000 × *g* for 15 min at 12 °C. The aqueous phase was transferred to a new tube. RNA was purified from this solution using the Agencourt RNAClean XP Kit (Beckman Coulter, A63987). Samples were then treated with TURBO DNase (Thermo Fisher, AM2238) and quality checked using a TapeStation 4150 (Agilent, G2992AA). For ribosomal RNA depletion with the 'do-it-yourself' method[61], 150–180 ng of total RNA was used. Library preparation for sequencing was carried out using NEBNext Ultra II Directional RNA Library Prep with Sample Purification Beads (NEB, E7765S). Sequencing was carried out at the Max Planck Genome Centre (Cologne, Germany) using an Illumina HiSeq3000 with 150 bp single reads, to obtain approximately 7 million reads per sample.

### Transcriptome analysis
The sequencing read files were imported into the software CLC Genomics Workbench v10.1.1 (Qiagen) and mapped to the *V. cholerae* reference genome (National Center for Biotechnology Information accession numbers, NC_002505.1 and NC_002506.1) using the 'RNA-Seq Analysis' function in the CLC software with standard parameters. Reads mapping to annotated coding sequences were counted, normalized (transcript per million, TPM) and transformed (log₂). Differential expression between the conditions was tested using the 'Differential Expression for RNA-Seq' command in the CLC software. Genes with a read count <10 in any condition were excluded from analysis. Genes with a fold change >2.0 and a false discovery rate (FDR)-adjusted *P* < 0.05 were defined as differentially expressed. Candidate genes were categorized by keyword enrichment using information imported from UniProt[62], Kyoto Encyclopedia of Genes and Genomes[63] and MicrobesOnline[64].

### Microscopy
Immediately after initiation of the flow of media through the microfluidic device, *V. cholerae* cells were imaged every 20 min or 30 min for up to 8 h. Imaging was performed with a Yokogawa CSU confocal spinning disk unit mounted on a Nikon Ti-E inverted microscope using a ×60 oil objective with numerical aperture 1.4 (Nikon) for measurements of the biofilm biovolume fraction, or a ×100 silicon oil objective with numerical aperture 1.35 (Olympus) for measurements of spatiotemporal fluorescent reporters. Fluorescent proteins were excited with a 488 nm laser (sfGFP and mNeonGreen) or a 552 nm laser (mRuby2, mRuby3 and Tag-RFP-T). The microscope hardware was controlled by NIS Elements (Nikon) or by Micro-Manager 2.0beta. Images were captured by an Andor iXon EMCCD camera, cooled to −70 °C. Images were acquired at low excitation light intensities with 90 ms exposure time while amplifying the readout using the EM-gain of the camera. A Nikon PFS hardware autofocus was used to correct focus drift. Image stacks were acquired at a spatial resolution of 63.2 nm in the *xy*-plane and 0.5 µm in the *z*-direction.

### Image analysis
All image analyses were performed with the software tool BiofilmQ v.1.0.1[15]. For biofilm biovolume quantification, the cells were distinguished from the background based on the constitutively expressed fluorescent protein. Before segmentation, a mean filter with the kernel size 5 px (*xy*), 3 px (*z*) was applied to the images to reduce the photon shot noise. To reduce the segmentation artefacts introduced by free floating cells around the biofilm, the floating cell suppression algorithm of BiofilmQ was used. The cellular signal was enhanced above the background using a top-hat filter with a disk-shaped kernel larger than the expected bacterial size (here 5.5 µm or 25 px), which was applied per *xy*-image in the *z*-stack. The segmentation was performed on the resulting images by an Otsu threshold multiplied with 0.2 to make sure all fluorescent objects were captured correctly. The discount factor (0.2) was kept constant for all image segmentations. Subsequently, the total biovolume and the ratio of the biovolume above 3 µm from the substrate divided by the total biovolume were calculated.

For the spatiotemporal quantification of fluorescent reporters, the biofilm biovolume was segmented as described above, and the images were then further segmented into cubes with a side length of approximately 1 µm using BiofilmQ. For each cube, the mean fluorescence intensity or the ratio of the two measured fluorescent channels was calculated. In addition, the distance of each cube to the interface of the biofilm with the surrounding liquid was calculated, with a resolution of ~1 µm. To construct kymographs for the spatiotemporal analysis of fluorescent reporters, the parameter values of all cubes with a similar distance to the interface of the biofilm with the surrounding liquid were averaged, resulting in a value of a pixel in the space–time heat maps shown in Fig. 4c,d.

### Statistical analysis and data presentation
Bar graphs were generated using Graphpad Prism v9, which was also used for performing all statistical tests indicated in figure captions. Three-dimensional rendering of confocal microscopy images of bacterial cells was performed using Paraview v.5.7[65], based on segmentation results and output from BiofilmQ[15].

## Reporting summary

Further information on research design is available in the Nature Portfolio Reporting Summary linked to this article.

## Data availability

Transcriptome data are available at the National Center for Biotechnology Information Gene Expression Omnibus under the accession number GSE216690. Raw image data are available via Zenodo at https://doi.org/10.5281/zenodo.14054836 (ref. 66). Source data are provided with this paper.

## Code availability

Source code of the Matlab script used to quantify the 3D biofilm biovolume is available in a Github repository: https://github.com/knutdrescher/biofilm-3D-biovolume, and a demo dataset for this code is available via Zenodo at https://doi.org/10.5281/zenodo.13166027 (ref. 67).

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

## Acknowledgements

We are grateful to M. Bayer, V. Potlog, S. Freier, K. Raveendran and L. Vidakovic for their help with strain construction; E. Jiménez Siebert for help with figure construction; F. Yildiz, K. Papenfort and M. Basler for important discussions; T. Glatter for proteomics analysis; H. Rohde for *Staphylococcus* strains; C. D. Nadell for advice at the early stage of the project; and P. Stallforth for advice regarding the signal identification. We are also grateful to L. Alvarez for generously providing us with the plasmid pET28-MltA$_{Vc}$. We thank the Max Planck Genome Centre for their sequencing service. This work in K.D.'s laboratory was supported by the European Union's Horizon 2020 research and innovation program through the European Research Council Starting Grant (716734), the Swiss National Science Foundation Consolidator Grant (TMCG-3_213801) and grants from the Deutsche Forschungsgemeinschaft (DR 982/5-1 and DR 982/6-1), Behrens-Weise-Stiftung, Minna-James-Heineman-Stiftung, Bundesministerium für Bildung und Forschung (TARGET-Biofilms) and the National Center of Competence in Research AntiResist funded by the Swiss National Science Foundation (grant number 51NF40_180541). In addition, this work was supported by grants to F.C. by The Swedish Research Council, The Knut and Alice Wallenberg Foundation, The Laboratory of Molecular Infection Medicine Sweden, Cancerfonden and The Kempe Foundation. The work by S.G. was supported by a grant from the Dr. Rolf M. Schwiete-Stiftung. Furthermore, this work was supported by fellowships from the International Max Planck Research School Marburg (to S.V., M.F.H., H.J.), the Alexander von Humboldt Foundation (to K.N.), the

Studienstiftung des deutschen Volkes (to H.J.) and the Joachim Herz Stiftung (to H.J.).

## Author contributions

S.V., P.K.S. and K.D. designed the project. S.V., D.S., D.K.H.R. and M.F.H. conducted experiments and data analysis. S.V., D.K.H.R., D.S. and M.F.H. generated strains. E.J. and H.J. wrote software for data analysis. K.N. prepared bacterial samples for RNA-seq. G.T. and F.C. performed digests of peptidoglycan and interpreted resulting data. S.G. provided wild isolate strains. S.V., D.S., D.K.H.R., P.K.S. and K.D. interpreted the data. K.D. supervised and coordinated the project. S.V. and K.D. wrote the paper with the help of all authors.

## Competing interests

The authors declare no competing interests.

## Additional information

**Extended data** is available for this paper at https://doi.org/10.1038/s41564-024-01886-5.

**Correspondence and requests for materials** should be addressed to Knut Drescher.

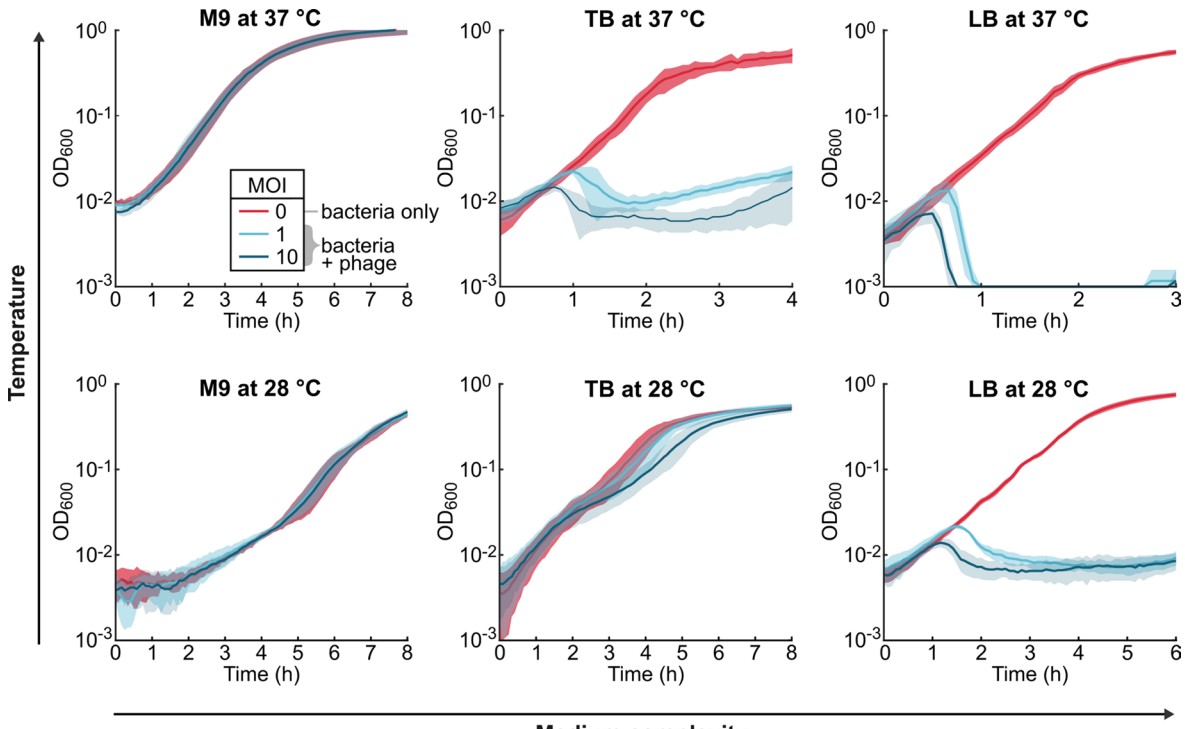

**Extended Data Fig. 1 | Vibriophage N4 infection of *V. cholerae* grown in different media and temperatures.** Each graph displays the infection dynamics in liquid shaking cultures, incubated at a particular temperature (28 °C or 37 °C) and in a particular medium (M9 minimal medium with 0.5% glucose, TB, LB). For each growth condition, phages were added to the bacterial suspension at time = 0 h at different multiplicity of infection (MOI, indicated by different line colours), and $OD_{600}$ was measured using a plate reader. Thick lines indicate the mean of $n$ = 3 biological replicates and the shaded regions represent the standard deviation. In M9 medium no drop in $OD_{600}$ was observed in the presence of phages, which indicates that there was no substantial phage-induced lysis. Phage-induced lysis is stronger in LB compared with TB, and stronger at 37 °C compared with 28 °C.

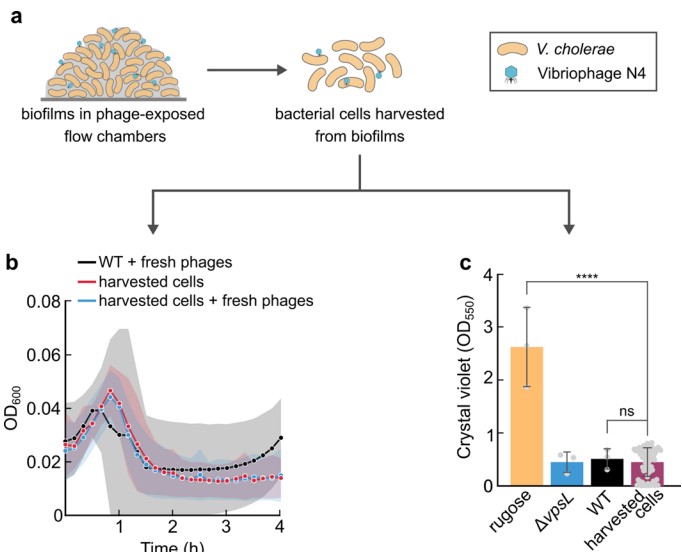

**Extended Data Fig. 2 | 3D biofilms that formed during phage exposure for 8 h are not composed of phage-resistant cells or matrix hyper-producing cells. a**, Schematic diagram of experimental workflow: Biofilms formed by *V. cholerae* after 8 h of phage exposure were harvested from flow chambers and separated into individual cells by strong vortexing, followed by two washes with LB to strongly reduce the concentration of phages. These harvested cells, which still contained a low concentration of adsorbed and intracellular phages, were used for the assays described in panels **b**, and **c**. **b**, Harvested *V. cholerae* cells were transferred to a 96-well plate and either fresh LB (red line) was added, or fresh LB containing purified Vibriophage N4 virions ($10^6$ PFU mL$^{-1}$, blue line) was added, followed by incubation at 37 °C with shaking for 4 h. As a control experiment, exponentially growing *V. cholerae* WT cells (not previously exposed to phages) were also inoculated with phages ($10^6$ PFU mL$^{-1}$, black line). These experiments show that most of the cells that were harvested from biofilms after 8 h of phage-exposure were still susceptible to phage infection.

Lines represent the mean of $n = 3$ biological replicates and shaded regions indicate standard deviations. **c**, Harvested *V. cholerae* cells were isolated as individual colonies and then cultured in 96-well plates to measure their level of biofilm matrix production using the crystal violet assay. Control strains: *V. cholerae* WT, Δ*vpsL* (lacking essential matrix component VPS), rugose strain (*vpvC*$^{W240R}$ allele, resulting in matrix hyper-production[68]). The matrix hyper-producing rugose strain displayed higher biofilm production than the WT and Δ*vpsL* strain, as described previously[68]. Harvested cells produced a crystal violet biofilm signal that was similar to the WT and Δ*vpsL*. Bars are mean values of $n = 3$ independent biological replicates for the rugose, WT and Δ*vpsL* strains, and $n = 84$ independent isolates that were harvested from the channels; points denote individual measurements, and error bars represent standard deviations. Statistical significances between the harvested isolates and the WT control and the rugose control were calculated using a two-sided Student's *t*-test: ns = not significant ($p = 0.7097$), and **** is $p < 0.0001$.

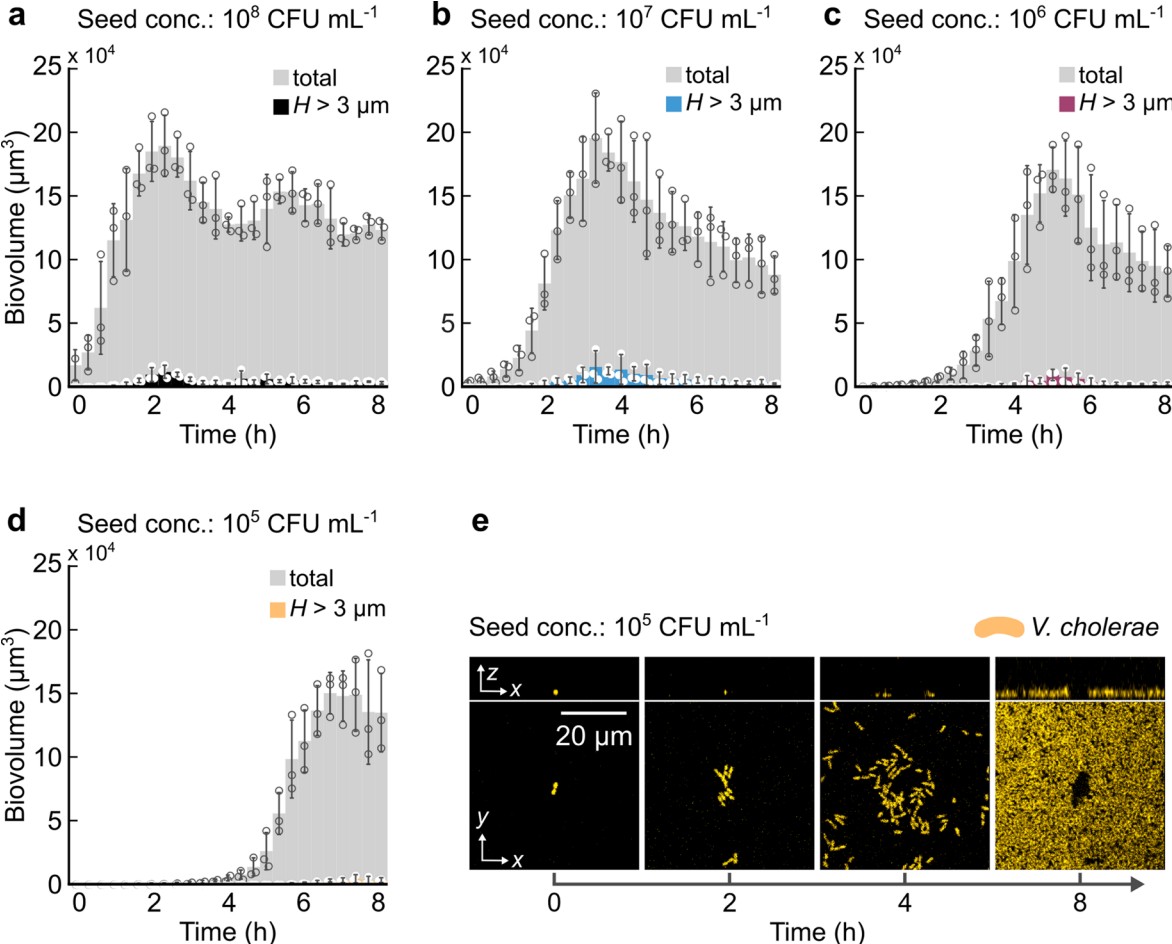

**Extended Data Fig. 3 | Reduction of the *V. cholerae* seeding cell density in flow chambers does not result in 3D biofilm formation in the absence of phages.** Experiments were performed at 37 °C in flow chambers through which LB medium was flowed after inoculating the channel with a bacterial seeding population of a given cell concentration, as described in the Methods section. **a-d**, Quantification of the total biovolume (grey bars) and biovolume with height $H > 3\,\mu m$ (coloured bars) of *V. cholerae* cells grown in microfluidic chambers with continuous flow without phages. Bacteria were diluted to different starting concentrations for seeding the flow channels: **a**, $10^8$ CFU mL$^{-1}$ (black; represents the same condition as Fig. 1c, unexposed); **b**, $10^7$ CFU mL$^{-1}$ (blue); **c**, $10^6$ CFU mL$^{-1}$ (purple), **d**, $10^5$ CFU mL$^{-1}$ (yellow). None of these different seeding concentrations resulted in substantial 3D biofilm formation, *that is* biovolume above $H > 3\,\mu m$. Bars are mean values with individual data points denoting $n = 3$ biological replicates and error bars indicate the standard deviation. **e**, Confocal image time series of *V. cholerae* cells (yellow, constitutively producing sfGFP), inoculated at an initial cell density of $10^5$ CFU mL$^{-1}$, grown in the absence of phages.

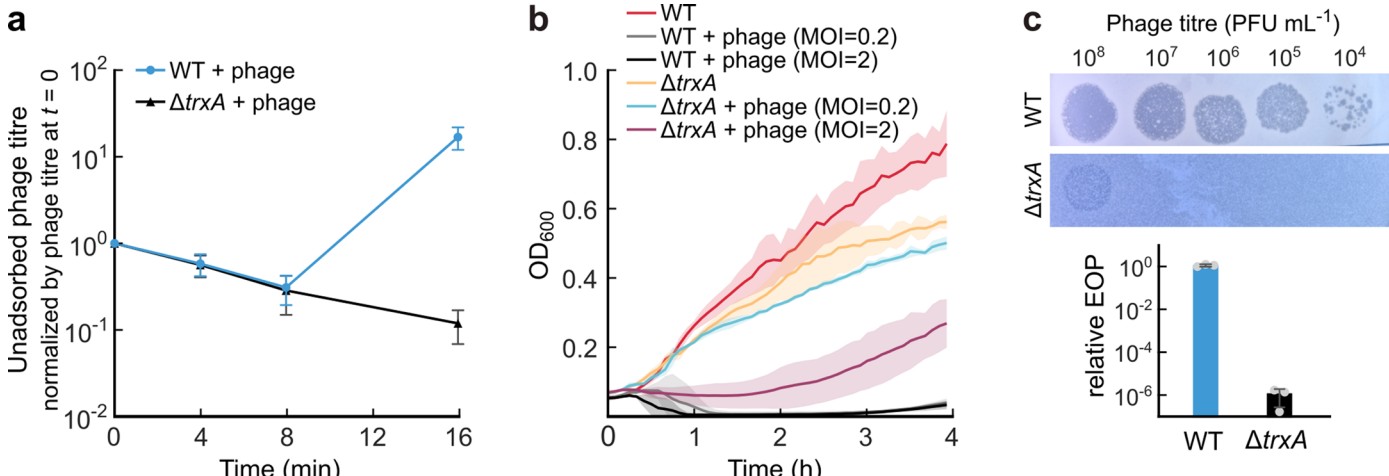

**Extended Data Fig. 4 | The *V. cholerae* Δ*trxA* mutant displays less lysis after phage exposure compared to the wild type. a**, Phage adsorption dynamics of *V. cholerae* WT and Δ*trxA* cells, following exposure to Vibriophage N4 (MOI = 0.001) at time = 0 min. Unadsorbed phage particles were enumerated by PFU assays and normalized by the initial phage titre at $t$ = 0. Phage adsorption to the bacterial cells was unaffected by the *trxA* deletion, but the concentration of progeny phages at 16 min after phage exposure is ~100x diminished in the Δ*trxA* strain, compared to the WT. Points represent the mean of 3 biological replicates and error bars indicate the standard deviation. **b**, Growth curves in liquid shaking cultures of the WT and Δ*trxA* strain, either unexposed to phages, or exposed to phages with MOI = 0.2 or MOI = 2 at time = 0 h. Growth of the Δ*trxA* strain was only weakly affected by phages at MOI = 0.2, and at MOI = 2 there was a growth

delay. In contrast, WT cells were strongly affected by phage exposure at both MOI levels. Lines represent the mean of 3 biological replicates and shaded regions indicate the standard deviation. **c**, The efficiency of plating (EOP) was assayed by spotting 10 μL of phage suspensions with different phage titres onto a lawn of bacterial cells on an LB agar plate, followed by incubation at 37 °C. Images were acquired using a stereomicroscope. The relative EOP was then calculated by dividing the plaque forming units (PFU) by the titre in a phage inoculation spot. The relative EOP measurements indicate that the Δ*trxA* strain displays severely attenuated phage-induced lysis compared to the WT. Bars are mean values, circles denote 3 biological replicates for each condition, and error bars indicate the standard deviation.

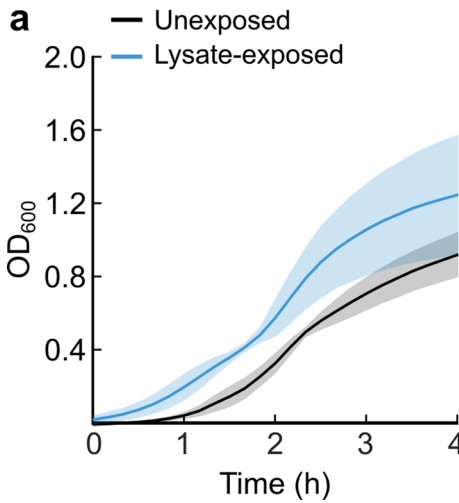

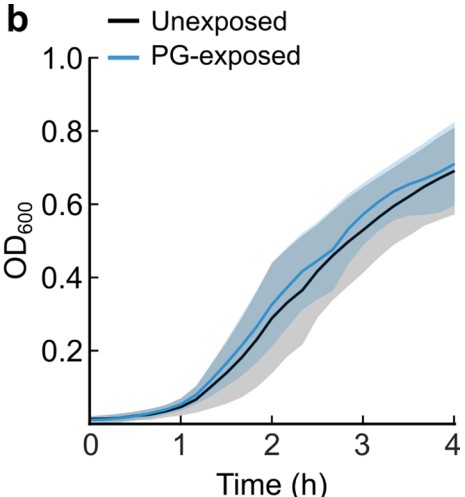

**Extended Data Fig. 5 | Growth curves of *V. cholerae* WT in the presence of cell lysate or peptidoglycan (PG).** Cells were inoculated at $OD_{600}$ = 0.04 and incubated in liquid LB under shaking conditions. Lines represent the mean of $n$ = 3 biological replicates and shaded regions indicate the standard deviation. **a**, Growth curves of *V. cholerae* WT either with (blue) or without (black) the addition of lysate (final concentration of $10^9$ lysed cells $mL^{-1}$). **b**, Growth curves of *V. cholerae* WT either with (blue) or without (black) the addition of PG (final concentration of 300 µg $mL^{-1}$).

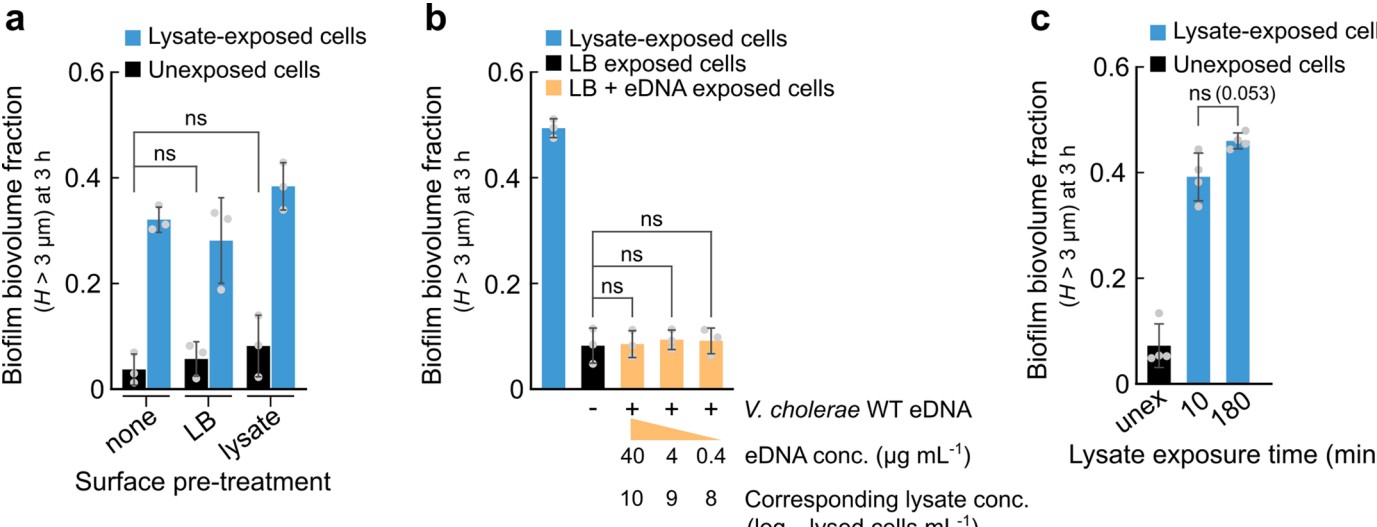

**Extended Data Fig. 6 | Lysate components do not serve as a scaffold for 3D biofilm formation. a,** Treating surfaces with lysate prior to inoculation of *V. cholerae* cells does not cause 3D biofilm formation in flow chambers. 3D biofilm formation of *V. cholerae* WT was measured while the cells were either exposed to lysate ($10^9$ lysed cells mL$^{-1}$) or unexposed to lysate (in this case: exposed to LB medium), for flow chambers that were treated with different conditions prior to inoculation of the bacterial cells. The flow chambers were pre-treated for 60 min with either LB, or lysate ($10^9$ lysed cells mL$^{-1}$), or no pre-treatment. Pre-treating the surface of the microfluidic chambers with lysate did not significantly change 3D biofilm formation. **b,** *V. cholerae* WT cells that were exposed to varying concentrations of extracellular DNA (eDNA, isolated from *V. cholerae* WT lysate) did not display biofilm formation, similar to the unexposed condition. **c,** Exposure of *V. cholerae* WT cells to lysate ($10^{10}$ lysed cells mL$^{-1}$) for only 10 min

followed by 170 min of exposure to medium without lysate, or exposure to lysate for 180 min induced similar levels of biofilm formation. For all measurements in this figure, biofilm formation was quantified by calculating the 3D biofilm biovolume fraction, which is the biovolume with height $H > 3$ μm divided by the total biovolume of the bacterial cells. Bars are mean values with points denoting $n = 3$ biological replicates for panels **a,b** and $n = 4$ biological replicates for panel **c**. Error bars indicate the standard deviation. For panel **a-b**, statistical significances were calculated using a one-way ANOVA with Bonferroni's correction (ns = not significant). In panel **a**, $p > 0.9999$ for both tests. In panel **b**, LB-exposed vs. 40 μg/mL eDNA-exposed yielded $p > 0.9999$, LB-exposed vs. 4 μg/mL eDNA-exposed yielded $p = 0.803$, LB-exposed vs. 0.4 μg/mL eDNA-exposed yielded $p = 0.941$. For panel **c**, statistical significance was calculated using a two-sided Student's *t* test, resulting in $p = 0.053$.

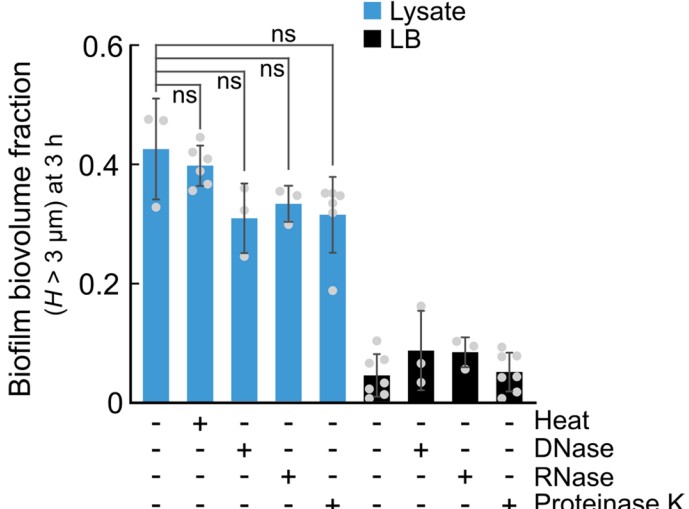

**Extended Data Fig. 7 | Heat, DNase, RNase, or proteinase treatment of the cell lysate did not reduce the 3D biofilm induction capacity of cell lysate.** *V. cholerae* WT cells in microfluidic flow chambers were exposed to lysate of *V. cholerae* WT cells (obtained by sonication, $10^{10}$ lysed cells $mL^{-1}$ in LB medium) which was subjected to different treatments: heat (80 °C for 20 min), DNase I (1 U $mL^{-1}$ at 37 °C for 30 min), RNase A (1 μg $mL^{-1}$ at 37 °C for 30 min), or proteinase K (20 μg $mL^{-1}$ at 37 °C for 60 min). As control conditions, LB medium that was subjected to the same treatments was flushed into flow chambers seeded with *V. cholerae* WT cells. The heat or enzyme treatments did not diminish the biofilm inducing capability of the lysate. Bars are mean values of *n* independent biological replicates, where *n* is as follows for the different conditions (bar1 corresponds to the left-most bar in the graph and bar9 corresponds to the right-most bar): $n_{bar1} = 3$, $n_{bar2} = 6$, $n_{bar3} = 3$, $n_{bar4} = 3$, $n_{bar5} = 6$, $n_{bar6} = 7$, $n_{bar7} = 3$, $n_{bar8} = 3$, $n_{bar9} = 7$. Circles indicate individual measurements, and error bars indicate the standard deviation. Statistical significances were calculated using a one-way ANOVA with Bonferroni's correction, with the following results: ns = not significant, where untreated lysate vs. heat-treated lysate yielded $p = 0.483$, untreated lysate vs. DNase-treated lysate yielded $p = 0.115$, untreated lysate vs. RNase-treated lysate yielded $p = 0.142$, untreated lysate vs. proteinase K-treated lysate yielded $p = 0.0617$.

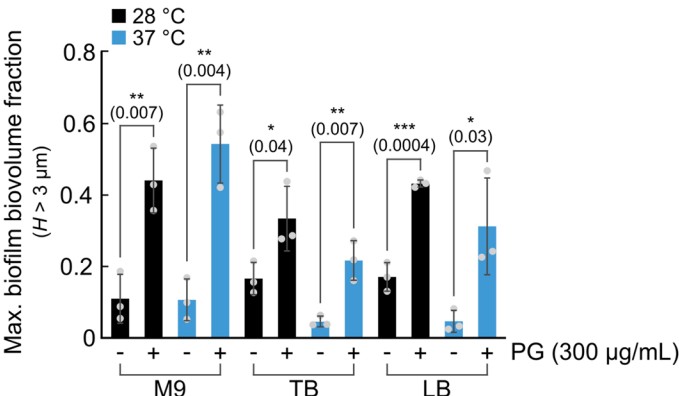

**Extended Data Fig. 8 | Exogenous peptidoglycan induces 3D biofilm formation of *V. cholerae* in different media and temperatures.** *V. cholerae* biofilm growth in our microfluidic system was measured in the presence (+) or absence (-) of 300 µg mL⁻¹ exogenous peptidoglycan (PG), at a particular temperature (28 °C or 37 °C) and in a particular liquid medium (M9 minimal medium with 0.5% glucose, TB, LB). As growth rates strongly differ between different media and temperatures, we measured the 3D biofilm biovolume fraction with height $H > 3$ µm in a given growth condition at the time of maximum biofilm height, which is a time that differed between different growth conditions. The time at which the maximum biovolume fraction at heights $H > 3$ µm occurs is as follows: for M9, $t = 7.5$-$13.5$ h at 28 °C and 9-11 h at 37 °C; for TB, $t = 3.5$-$4.5$ h at 28 °C and 2.5-5.0 h at 37 °C; for LB, $t = 2.5$-$4.5$ h at 28 °C and 2-3 h at 37 °C). In each growth condition, the biofilm biovolume was measured at the same time for the PG-exposed and unexposed condition. In all growth conditions, PG-exposure resulted in a statistically significant enhancement of 3D biofilm formation. Bars indicate the mean of $n = 3$ biological replicates, error bars indicate the standard deviation and individual data points are shown. Statistical significances were calculated using a two-sided Student's *t*-test; * = $p < 0.1$; ** = $p < 0.01$; *** = $p < 0.001$. Exact *p*-values are given in brackets in the figure.

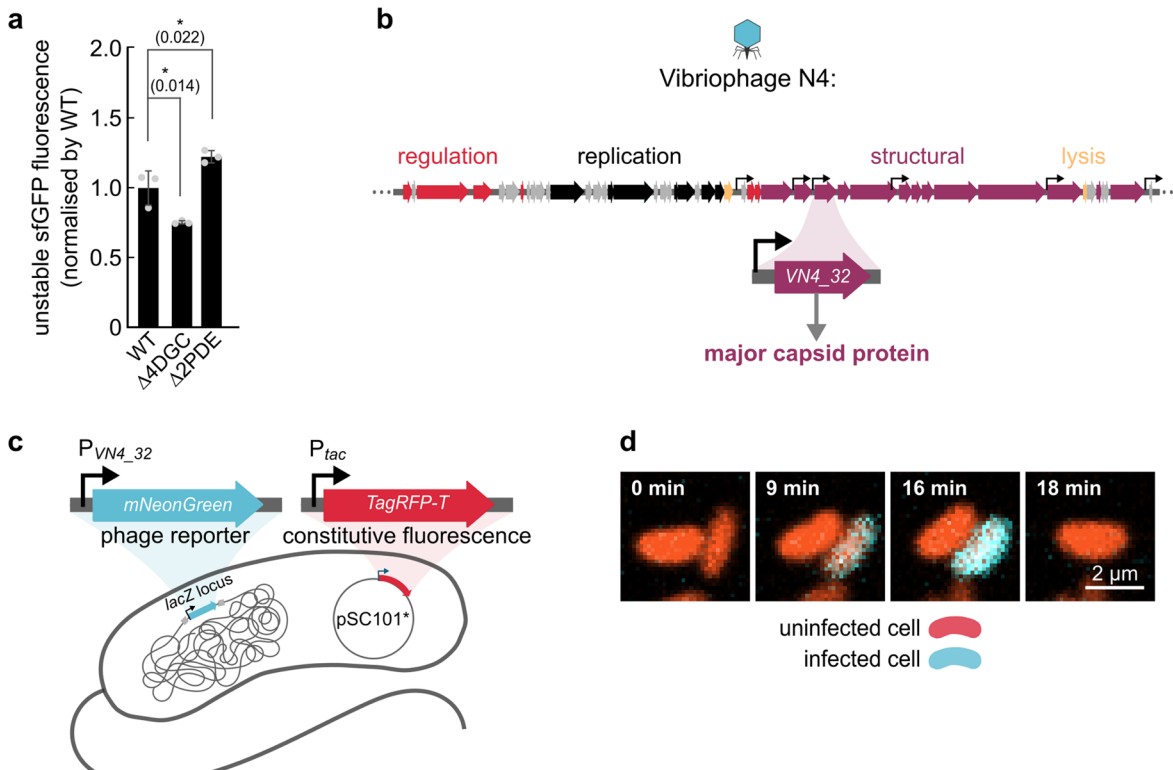

**Extended Data Fig. 9 | Fluorescent protein-based reporters for c-di-GMP level and for Vibriophage N4 infection in *V. cholerae* cells. a**, Calibration results for the c-di-GMP reporter. The reporter is based on the triple-tandem riboswitch Bc3-5[69], which permits the transcription of the *sfGFP-LAA* gene, coding for an unstable superfolder-GFP with the LAA degradation tag[53]. The *bc3-5-sfGFP-LAA* fragment was cloned into a low copy-number plasmid (pSC101*) harboured in *V. cholerae*. The bar graph shows the quantification of the unstable-sfGFP fluorescent intensity levels in microscopy images, normalized by the mean of the WT level, for three different strains that are known to have different levels of c-di-GMP[70], which were grown in liquid shaking culture until $OD_{600} = 0.4$. The Δ4DGC strain lacks four diguanylate cyclases (Δ*cdgD*Δ*cdgK*Δ*cdgH*Δ*cdgL*), which are proteins that can produce c-di-GMP. The Δ2PDE strain lacks two phosophodiesterases (Δ*rocS*Δ*cdgJ*), which are proteins that can degrade c-di-GMP. The cellular c-di-GMP levels were expected to be intermediate for the WT, low for the Δ4DCG mutant, and high for the Δ2PDE mutant[70,71]. The fluorescence levels of the unstable sfGFP correspond qualitatively to the expected c-di-GMP levels. Bars are mean values with points denoting $n = 3$ biological replicates and error bars indicate the standard deviation. Statistical significances were calculated using a one-way ANOVA with Bonferroni's correction, yielding *p*-values that are indicated in brackets in the graph underneath the * symbol. **b**, To construct the fluorescent protein-based reporter for Vibriophage N4 infection in *V. cholerae* cells, the promoter of the gene *VN4_32* (encoding the major capsid protein) was identified on the Vibriophage N4 genome using PHIRE[72]. **c**, Schematic drawing of the phage infection reporter system: The phage promoter $P_{VN4\_32}$ was fused to *mNeonGreen* (cyan) and inserted at the *lacZ* locus on the *V. cholerae* chromosome. Constitutive fluorescence was achieved by engineering a $P_{tac}$-*TagRFP-T* (red) construct on a low-copy plasmid (pSC101*). **d**, Confocal microscopy image time series showing the production of mNeonGreen (cyan) during phage infection, followed by cell lysis. All cells produce TagRFP-T (red).

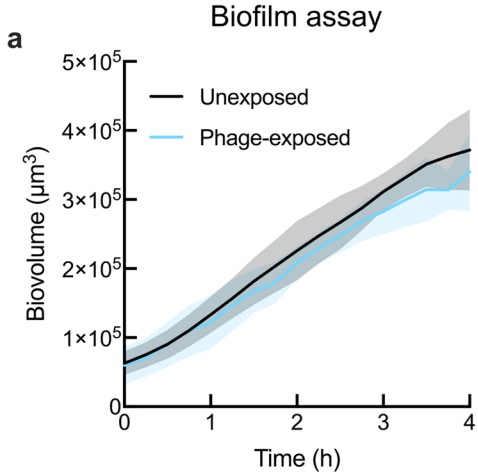

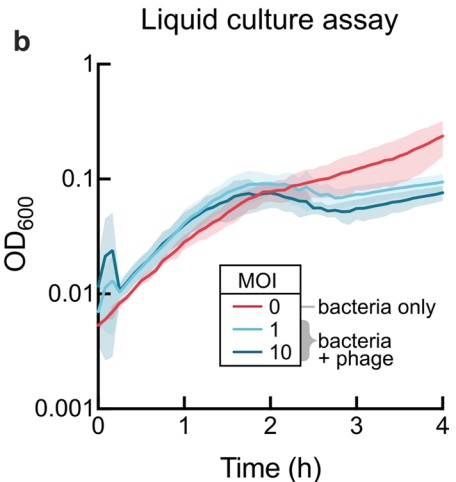

**Extended Data Fig. 10 | Growth curves of *V. cholerae* rugose strain during phage exposure in biofilm and liquid culture conditions.** The *V. cholerae* rugose strain (KDV1502) carries the *vpvC^W240R^* allele, resulting in matrix hyper-production[1] and strong 3D biofilm formation in LB medium, even in the absence of exogenous peptidoglycan. **a**, Biofilms of the rugose strain were grown in flow chambers in LB medium at 37 °C for 3 h, resulting in 3D biofilm colonies. Then, at time $t = 0$ h, the inflowing medium was exchanged to LB containing purified phages ($10^7$ PFU mL$^{-1}$), or LB containing no phages, and the biofilm biovolume was monitored using confocal microscopy and analysed using BiofilmQ. The presence of phages only had a small impact on the biofilm biovolume, indicating that the biofilm population is largely protected from phages. Thick lines indicate the mean of $n = 3$ biological replicates and the shaded regions represent the standard deviation. **b**, Liquid cultures of the rugose strain were grown in LB medium at 37 °C under shaking conditions. When back-diluting the pre-culture to $OD_{600} = 0.01$ at time $t = 0$ h, purified phages were added and the $OD_{600}$ was monitored using a plate reader. The drop in $OD_{600}$ within the first hour of co-incubation with phages indicates that the rugose strain is susceptible to phage infection (see also the phage adsorption and phage release dynamics for the wild type in Extended Data Fig. 4a, for comparison of timescales). The second drop in $OD_{600}$ around 2 h may result from a second wave of phage infection. Thick lines indicate the mean of $n = 3$ biological replicates and the shaded regions represent the standard deviation.

# Reporting Summary

## Statistics

For all statistical analyses, confirm that the following items are present in the figure legend, table legend, main text, or Methods section.

| n/a | Confirmed | |
|---|---|---|
| ☐ | ☒ | The exact sample size (*n*) for each experimental group/condition, given as a discrete number and unit of measurement |
| ☐ | ☒ | A statement on whether measurements were taken from distinct samples or whether the same sample was measured repeatedly |
| ☐ | ☒ | The statistical test(s) used AND whether they are one- or two-sided<br>*Only common tests should be described solely by name; describe more complex techniques in the Methods section.* |
| ☒ | ☐ | A description of all covariates tested |
| ☐ | ☒ | A description of any assumptions or corrections, such as tests of normality and adjustment for multiple comparisons |
| ☐ | ☒ | A full description of the statistical parameters including central tendency (e.g. means) or other basic estimates (e.g. regression coefficient) AND variation (e.g. standard deviation) or associated estimates of uncertainty (e.g. confidence intervals) |
| ☐ | ☒ | For null hypothesis testing, the test statistic (e.g. *F*, *t*, *r*) with confidence intervals, effect sizes, degrees of freedom and *P* value noted<br>*Give P values as exact values whenever suitable.* |
| ☒ | ☐ | For Bayesian analysis, information on the choice of priors and Markov chain Monte Carlo settings |
| ☒ | ☐ | For hierarchical and complex designs, identification of the appropriate level for tests and full reporting of outcomes |
| ☒ | ☐ | Estimates of effect sizes (e.g. Cohen's *d*, Pearson's *r*), indicating how they were calculated |

*Our web collection on statistics for biologists contains articles on many of the points above.*

## Software and code

Policy information about availability of computer code

| | |
|---|---|
| Data collection | Nikon NIS Elements Advanced Research 4.5 software and Micro-Manager 2.0beta were used to control microscopes for image acquisition. |
| Data analysis | BiofilmQ v0.2.2 was used to analyse biofilm images: https://drescherlab.org/data/biofilmQ/ .<br>Bar graphs were generated using Graphpad Prism v9, which was also used for performing all statistical tests indicated in figure captions.<br>Three-dimensional rendering of confocal microscopy images of bacterial cells was performed using Paraview v5.10.1, based on segmentation results and output from BiofilmQ.<br>CLC Genomics Workbench v10.1.1 (Qiagen) was used to process RNA-seq data obtained by Illumina sequencing.<br>Source code of the Matlab script we used to quantify the 3D biofilm biovolume is available in a Github repository: https://github.com/knutdrescher/biofilm-3D-biovolume. |

For manuscripts utilizing custom algorithms or software that are central to the research but not yet described in published literature, software must be made available to editors and reviewers. We strongly encourage code deposition in a community repository (e.g. GitHub). See the Nature Portfolio guidelines for submitting code & software for further information.

## Data

Policy information about [availability of data](availability of data)

All manuscripts must include a [data availability statement](data availability statement). This statement should provide the following information, where applicable:

- Accession codes, unique identifiers, or web links for publicly available datasets
- A description of any restrictions on data availability
- For clinical datasets or third party data, please ensure that the statement adheres to our [policy](policy)

Transcriptome data are available at the National Center for Biotechnology Information Gene Expression Omnibus under the accession number GSE216690 (https://www.ncbi.nlm.nih.gov/geo/query/acc.cgi?acc=GSE216690).
Image data are available on the Zenodo repository (DOI: 10.5281/zenodo.14054836).
Processed data used in this study are available as Source Data together with the manuscript.

## Human research participants

Policy information about [studies involving human research participants and Sex and Gender in Research.](studies involving human research participants and Sex and Gender in Research.)

| | |
|---|---|
| Reporting on sex and gender | n/a |
| Population characteristics | n/a |
| Recruitment | n/a |
| Ethics oversight | n/a |

Note that full information on the approval of the study protocol must also be provided in the manuscript.

# Field-specific reporting

Please select the one below that is the best fit for your research. If you are not sure, read the appropriate sections before making your selection.

☒ Life sciences          ☐ Behavioural & social sciences          ☐ Ecological, evolutionary & environmental sciences

For a reference copy of the document with all sections, see [nature.com/documents/nr-reporting-summary-flat.pdf](nature.com/documents/nr-reporting-summary-flat.pdf)

# Life sciences study design

All studies must disclose on these points even when the disclosure is negative.

| | |
|---|---|
| Sample size | The number (n) of independent replicate experiments that were performed for each experiment was determined by the minimum number of biological replicates required for good data distribution and statistics. For most experiments reported in this study n=3, which is a generally accepted number of independent replicates in the research field. However, for some assays n>3 independent replicate were acquired when the assay cold be done in higher throughput. The exact sample size for each experiment is indicated in the figure captions. |
| Data exclusions | No data were excluded. |
| Replication | Each experiment was replicated n times (and n is given in each figure for each experiment). Although the exact quantitative results differ between replicates, the qualitative results were the same, so that it is reasonable to state that the "replication was successful". |
| Randomization | There were many bacterial cells within each of the n replicates. Because of the large sample size for each replicate, a representative number of samples were collected for each replicate. There was no allocation of samples into experimental groups, beyond conducting independent biological replicates. |
| Blinding | Blinding of group allocation is irrelevant to our data analysis, because there was no allocation to experimental groups, beyond collecting n replicates, all of which were analyzed by software equally. |

# Reporting for specific materials, systems and methods

We require information from authors about some types of materials, experimental systems and methods used in many studies. Here, indicate whether each material, system or method listed is relevant to your study. If you are not sure if a list item applies to your research, read the appropriate section before selecting a response.

## Materials & experimental systems

| n/a | Involved in the study |
|-----|----------------------|
| ☒ | Antibodies |
| ☒ | Eukaryotic cell lines |
| ☒ | Palaeontology and archaeology |
| ☒ | Animals and other organisms |
| ☒ | Clinical data |
| ☒ | Dual use research of concern |

## Methods

| n/a | Involved in the study |
|-----|----------------------|
| ☒ | ChIP-seq |
| ☒ | Flow cytometry |
| ☒ | MRI-based neuroimaging |

