## [Peer Review File · Nature Microbiology]

Bacteria use exogenous peptidoglycan as a danger signal to trigger biofilm formation

Corresponding Author: Professor Knut Drescher

Version 0:

Reviewer comments:

Reviewer #1

(Remarks to the Author)

In this work, the authors observed that the short-term exposure of biofilms of the susceptible *Vibrio cholerae* El Tor strain C6706 to the lytic phage N4 caused the formation of elevated structured three dimensional biofilms with a distinct transcriptional profile and elevated concentration of the second messenger cyclic di-GMP compared to otherwise flat unstructured biofilm formation when grown in LB medium at body temperature. Subsequently, the compound(s) derived from cell lysate that trigger elevated biofilm formation were identified as peptidoglycan. Peptidoglycan was subsequently identified as a broad stimulator of biofilm formation in Gram-negative and Gram-positive bacteria equally as peptidoglycan from different species was able to induce elevated biofilm formation.

This work describes a novel twist in the delicate interplay between a lytic phage and biofilm-forming bacteria with the identification of the components that direct the interactions. The work has been well performed. I am wondering, however, what arguments/interpretation do the authors have with respect to the observed elevated biofilm formation upon phage exposure in rich medium at body temperature? Was a similar phenomenon also observed at lower temperature and/or using minimal medium?

General comments

Introduction: Should contain relevant information about *Vibrio cholerae* relevant phages and/or phage/biofilm interactions.

I.173: It seems that the peptidoglycan fragments that induce biofilm formation are quite large and thereby unique in their capability to conduct a biological reaction. Were these compounds digested with peptidoglycan processing enzymes other than lysozyme? Such a treatment should in any case abolish the biofilm inducing capability if the length is indeed a determinative factor.

I.185: Any speculations about the nature of alternative compound(s) that trigger elevated biofilm formation in spheroplasts?

I.291: Not sure that this is directly comparable as the peptidoglycan fragments recognized by NOD1 and NOD2 receptors are considered to be the subunits of peptidoglycan. There should be some speculation about the unique nature of the signal as well as the signaling mechanism.

Extended Data Figure 1c: This Figure is not clear with bacterial colonies poorly recognized, if at all.

Reviewer #2

(Remarks to the Author)

In this very interesting manuscript, the authors propose a new mechanism for bacterial community sensing of damage to neighboring cells. Bacteria exposed to phages, which cause cell lysis, or to cell lysates formed 3D biofilms, as opposed to unexposed bacteria, which primarily grew in a 2D monolayer. Bacteria in biofilms were found to be resistant to phage predation. The same phenomenon was observed in bacteria exposed to peptidoglycan alone, suggesting it is the component being sensed. If peptidoglycan was damaged by lysozyme this phenomenon was no longer observed. Exposure to peptidoglycan induced changes in gene expression that included increases in c-di-GMP, an important bacterial second messenger, and increases in biofilm matrix production, which makes sense given the obvious changes in biofilm production on a larger scale. Remarkably, this effect of increased biofilm formation in response to peptidoglycan exposure was observed in several bacterial species and did not require peptidoglycan from any particular species, suggesting it is a very common mechanism for protecting a community.

The manuscript is very well written. I have only minor suggestions for improvement.

Specific points:

- 1) Congratulations to the authors for very interesting and well-described work!
- 2) Fig 4: it would be beneficial to highlight which virulence genes are upregulated in response to peptidoglycan as this could be a major factor during an infection (appears to be TCP production and MakA toxin. It is curious that TcpA is apparently not upregulated but other parts of the operon are.)
- 3) Fig 5: Are biofilms formed by another mechanism also resistant to phage predation or is this specific to peptidoglycan sensing? This control would add to the conclusions.

Decision Letter:

16th December 2022

Dear Professor Drescher,

Thank you for your patience while your manuscript "Bacteria use exogenous peptidoglycan as a danger signal to trigger protective biofilm formation" was under peer-review at Nature Microbiology. It has now been seen by 2 referees, whose expertise and comments you will find at the end of this email. I'm happy to share that the referees are, overall, extremely positive about your work. You will see from their comments below that they agree about the interest and potential importance of your conclusions, however they do raise some important points. We are very interested in the possibility of publishing your study in Nature Microbiology, but would like to consider your response to these concerns in the form of a revised manuscript before we make a final decision on publication.

In particular, Reviewer #1 raises a question about whether or not the observed phenomenon would occur at lower temperatures or in different conditions. From an editorial perspective, we agree that addressing this question might increase the breadth of impact of your work. The rest of the referees' reports are clear and the remaining issues should be straightforward to address.

If you have not done so already please begin to revise your manuscript so that it conforms to our Article format instructions at <http://www.nature.com/nmicrobiol/info/final-submission/>

The usual length limit for a Nature Microbiology Article is six display items (figures or tables) and 3,000 words. We have some flexibility, and can allow a revised manuscript at 3,500 words, but please consider this a firm upper limit. There is a trade-off of ~250 words per display item, so if you need more space, you could move a Figure or Table to Supplementary Information.

Some reduction could be achieved by focusing any introductory material and moving it to the start of your opening 'bold' paragraph, whose function is to outline the background to your work, describe in a sentence your new observations, and explain your main conclusions. The discussion should also be limited. Methods should be described in a separate section following the discussion, we do not place a word limit on Methods.

Nature Microbiology titles should give a sense of the main new findings of a manuscript, and should not contain punctuation. Please keep in mind that we strongly discourage active verbs in titles, and that they should ideally fit within 90 characters each (including spaces).

Please include a data availability statement as a separate section after Methods but before references, under the heading "Data Availability". This section should inform readers about the availability of the data used to support the conclusions of your study. This information includes accession codes to public repositories (data banks for protein, DNA or RNA sequences, microarray, proteomics data etc...), references to source data published alongside the paper, unique identifiers such as URLs to data repository entries, or data set DOIs, and any other statement about data availability. At a minimum, you should include the following statement: "The data that support the findings of this study are available from the corresponding author upon request", mentioning any restrictions on availability. If DOIs are provided, we also strongly encourage including these in the Reference list (authors, title, publisher (repository name), identifier, year). For more guidance on how to write this section please see: <http://www.nature.com/authors/policies/data/data-availability-statements-data-citations.pdf>

To improve the accessibility of your paper to readers from other research areas, please pay particular attention to the wording of the paper's opening bold paragraph, which serves both as an introduction and as a brief, non-technical summary in about 150 words. If, however, you require one or two extra sentences to explain your work clearly, please include them even if the paragraph is over-length as a result. The opening paragraph should not contain references. Because scientists from other sub-disciplines will be interested in your results and their implications, it is important to explain essential but specialised terms concisely. We suggest you show your summary paragraph to colleagues in other fields to uncover any problematic concepts.

If your paper is accepted for publication, we will edit your display items electronically so they conform to our house style and will reproduce clearly in print. If necessary, we will re-size figures to fit single or double column width. If your figures contain several parts, the parts should form a neat rectangle when assembled. Choosing the right electronic format at this stage will speed up the processing of your paper and give the best possible results in print. We would like the figures to be supplied as vector files - EPS, PDF, AI or postscript (PS) file formats (not raster or bitmap files), preferably generated with vector-graphics software (Adobe Illustrator for example). Please try to ensure that all figures are non-flattened and fully editable. All images should be at least 300 dpi resolution (when figures are scaled to approximately the size that they are to be printed at) and in RGB colour format. Please do not submit Jpeg or flattened TIFF files. Please see also 'Guidelines for Electronic Submission of Figures' at the end of this letter for further detail.

Figure legends must provide a brief description of the figure and the symbols used, within 350 words, including definitions of any error bars employed in the figures.

Please include a statement before the acknowledgements naming the author to whom correspondence and requests for materials should be addressed.

Finally, we require authors to include a statement of their individual contributions to the paper -- such as experimental work, project planning, data analysis, etc. -- immediately after the acknowledgements. The statement should be short, and refer to authors by their initials. For details please see the Authorship section of our joint Editorial policies at http://www.nature.com/authors/editorial_policies/authorship.html

* include a point-by-point response to any editorial suggestions and to our referees. Please include your response to the editorial suggestions in your cover letter, and please upload your response to the referees as a separate document.

* ensure it complies with our format requirements for Letters as set out in our guide to authors at www.nature.com/nmicrobiol/info/gta/

* state in a cover note the length of the text, methods and legends; the number of references; number and estimated final size of figures and tables

* resubmit electronically if possible using the link below to access your home page:

Link Redacted

*This url links to your confidential homepage and associated information about manuscripts you may have submitted or be reviewing for us. If you wish to forward this e-mail to co-authors, please delete this link to your homepage first.

Please ensure that all correspondence is marked with your Nature Microbiology reference number in the subject line.

Nature Microbiology is committed to improving transparency in authorship. As part of our efforts in this direction, we are now requesting that all authors identified as 'corresponding author' on published papers create and link their Open Researcher and Contributor Identifier (ORCID) with their account on the Manuscript Tracking System (MTS), prior to acceptance. This applies to primary research papers only. ORCID helps the scientific community achieve unambiguous attribution of all scholarly contributions. You can create and link your ORCID from the home page of the MTS by clicking on 'Modify my Springer Nature account'. For more information please visit www.springernature.com/orcid.

We hope to receive your revised paper within three weeks. If you cannot send it within this time, please let us know.

Yours sincerely,

Reviewer Expertise:

Referee #1: biofilm biology, bacterial signaling

Referee #2: Vibrio ecology, host-pathogen interactions, bacterial stress response

Reviewers Comments:

Reviewer #1 (Remarks to the Author):

In this work, the authors observed that the short-term exposure of biofilms of the susceptible *Vibrio cholerae* El Tor strain C6706 to the lytic phage N4 caused the formation of elevated structured three dimensional biofilms with a distinct transcriptional profile and elevated concentration of the second messenger cyclic di-GMP compared to otherwise flat unstructured biofilm formation when grown in LB medium at body temperature. Subsequently, the compound(s) derived from cell lysate that trigger elevated biofilm formation were identified as peptidoglycan. Peptidoglycan was subsequently identified as a broad stimulator of biofilm formation in Gram-negative and Gram-positive bacteria equally as peptidoglycan from different species was able to induce elevated biofilm formation.

This work describes a novel twist in the delicate interplay between a lytic phage and biofilm-forming bacteria with the identification of the components that direct the interactions. The work has been well performed. I am wondering, however, what arguments/interpretation do the authors have with respect to the observed elevated biofilm formation upon phage exposure in rich medium at body temperature? Was a similar phenomenon also observed at lower temperature and/or using minimal medium?

General comments

Introduction: Should contain relevant information about *Vibrio cholerae* relevant phages and/or phage/biofilm interactions.

I.173: It seems that the peptidoglycan fragments that induce biofilm formation are quite large and thereby unique in their capability to conduct a biological reaction. Were these compounds digested with peptidoglycan processing enzymes other than lysozyme? Such a treatment should in any case abolish the biofilm inducing capability if the length is indeed a determinative factor.

I.185: Any speculations about the nature of alternative compound(s) that trigger elevated biofilm formation in spheroplasts?

I.291: Not sure that this is directly comparable as the peptidoglycan fragments recognized by NOD1 and NOD2 receptors are considered to be the subunits of peptidoglycan. There should be some speculation about the unique nature of the signal as well as the signaling mechanism.

Extended Data Figure 1c: This Figure is not clear with bacterial colonies poorly recognized, if at all.

Reviewer #2 (Remarks to the Author):

In this very interesting manuscript, the authors propose a new mechanism for bacterial community sensing of damage to neighboring cells. Bacteria exposed to phages, which cause cell lysis, or to cell lysates formed 3D biofilms, as opposed to unexposed bacteria, which primarily grew in a 2D monolayer. Bacteria in biofilms were found to be resistant to phage predation. The same phenomenon was observed in bacteria exposed to peptidoglycan alone, suggesting it is the component being sensed. If peptidoglycan was damaged by lysozyme this phenomenon was no longer observed. Exposure to peptidoglycan induced changes in gene expression that included increases in c-di-GMP, an important bacterial second messenger, and increases in biofilm matrix production, which makes sense given the obvious changes in biofilm production on a larger scale. Remarkably, this effect of increased biofilm formation in response to peptidoglycan exposure was observed in several bacterial species and did not require peptidoglycan from any particular species, suggesting it is a very common mechanism for protecting a community.

The manuscript is very well written. I have only minor suggestions for improvement.

Specific points:

- 1) Congratulations to the authors for very interesting and well-described work!
- 2) Fig 4: it would be beneficial to highlight which virulence genes are upregulated in response to peptidoglycan as this could be a major factor during an infection (appears to be TCP production and MakA toxin. It is curious that TcpA is apparently not upregulated but other parts of the operon are.)
- 3) Fig 5: Are biofilms formed by another mechanism also resistant to phage predation or is this specific to peptidoglycan sensing? This control would add to the conclusions.

Version 1:

Reviewer comments:

Reviewer #1

(Remarks to the Author)

This is the revised version of a manuscript previously submitted to Nature Microbiology. The authors have made great efforts and the manuscript has greatly improved.

I have only some comments to give on this revised version

1. In the title, abstract, result section and in the headline of figure legends, the authors mention enhanced biofilm formation. To avoid misunderstanding that needs to be changed to enhanced 3D biofilm formation.
2. In the extended Figure 10 the authors interpret the peak and drop during the first hour of phage infection as susceptibility of bacteria to the phage. Is the infection process that fast (within 15 min) considering that phage absorption upon infection initiation can already take much longer. What about the (slight) drop in OD after 2 h?
3. It might be informative to add a scheme with the peptidoglycan fragments created by the different enzymatic digests.

(Remarks on code availability)

Reviewer #2

(Remarks to the Author)

Nothing to add

(Remarks on code availability)

Reviewer #3

(Remarks to the Author)

In the work "Bacteria use exogenous peptidoglycan (PG) a danger signal to trigger biofilm formation" the authors show that exposure of bacteria to peptidoglycan leads to broad changes in gene expression resulting in taller, uniquely structured biofilms. Additionally the authors show that the source of this exogenous peptidoglycan can be from other bacterial species or kin that have undergone lytic phage infection. The authors have gone above-and-beyond in characterizing infections by the T7-like vibriophage N4 in their system. Biofilms and phage infections have been extensively studied in *V. cholerae* and this work begins to further show how the biofilm growth state is influenced by and influences interactions with phages.

Figure 1 beautifully shows a changes in biofilm architecture and volume when *V. cholerae* are exposed to the vibriophage N4. Extended exposure of *V. cholerae* to phages similar to N4 usually selects for phage receptor mutants, however in the context of these experiments the bacteria appear to remain susceptible to the phage. Mutation of the receptor(s) can often be deleterious in the environment of the host and often impact biofilm formation. If the unique biofilm structures induced by peptidoglycan sensing are a protective method that allows bacteria to avoid receptor mutations then it may be worth the authors commenting on. If nothing else, it may be helpful for the authors to speculate as to why they did not observe receptor mutations.

Experiments with the *trxA* strain were clever and it is clear a significant amount of effort went into the variety of phage related experiments in this work.

Phages encode diverse lysis cassettes which can include multiple different PG targeting enzymes, including endopeptidases, amidases, and lytic transglycosylases. Figure 3 shows that digestion by such enzymes reduces the observed biofilm phenotype upon exposure to PG, thereby suggesting that phages suppress this signaling even as they induce it. I was wondering if the authors had tested whether phages with disparate lysis cassettes still induced this biofilm phenotype?

Figure 4 utilizes different reporters to show transcriptional responses to peptidoglycan. I find this visualization to be an excellent way to convey a large amount of data. Changes after PG exposure for both the c-di-GMP and *vsp-I* reporter appear to occur on the timescale of hours and are spatially distributed throughout the biofilm. The infection cycle of the vibriophage N4 on *V. cholerae* C6706 in these conditions occurs in minutes and c-di-GMP signaling is often used for rapid responses to environmental changes. Do the authors have any insight on to how their reporter data temporally fits within this system?

Additionally either phage or peptidoglycan exposure should primarily occur on the perimeter of the biofilm, I was wondering if the authors could speak to how the signal would be propagated throughout the biofilm or if the changes observed are primarily due to density dependent changes in expression of the reporters.

It is interesting that the anti-phage CBASS system, CapV-DncV appears upregulated upon exposure to peptidoglycan. Do the authors think that this autolysis pathway could be involved in the restricting of biofilms after sensing peptidoglycan? Moreover, these genes, as well as many of the differentially regulated genes fall under regulation of quorum sensing genes such as HapR which itself appears to be differentially regulated by peptidoglycan exposure. The authors rule out cell density as an initiator of their biofilm phenotype, but do not comment on how the altered architecture may itself alter localized cell density and alteration of the quorum state of biofilm cells.

This phenomenon occurs throughout gram negative and gram-positive bacteria suggesting a conserved, or convergently evolved mechanism for exogenous peptidoglycan sensing and signal transduction. It makes sense that bacteria will have evolved a mechanism to generally use peptidoglycan as a "danger signal" as many different threats, such as the host, environmental changes, or antibiotics can cause lysis and release of peptidoglycan. Phage generally only infect a limited host range of closely related bacteria. Therefore in a complex environment, sensing a such broad signal might not be relevant to a

nearby unrelated and non-susceptible bacterium. I was wondering if the authors could speak to why they believe phages to be a primary source for this signal and the environmental context where such a biofilm response to phage infection may prove beneficial.

(Remarks on code availability)

The code was readily available with clear installation instructions and a practice dataset. However, I did not run the code on the practice data so I cannot speak to how smoothly it runs.

Decision Letter:

Our ref: NMICROBIOL-22102621A

26th September 2024

Dear Dr. Drescher,

Thank you for submitting your revised manuscript "Bacteria use exogenous peptidoglycan as a danger signal to trigger biofilm formation" (NMICROBIOL-22102621A). It has now been seen by the original referees and their comments are below. I am sorry that this has taken quite a long journey, but I am thrilled to finally be able to share good news with you: the reviewers find that the paper has improved in revision, and therefore we'll be happy in principle to publish it in Nature Microbiology, pending minor revisions to satisfy the referees' final requests and to comply with our editorial and formatting guidelines.

Thank you again for your interest in Nature Microbiology Please do not hesitate to contact me if you have any questions.

Sincerely,

Reviewer #1 (Remarks to the Author):

This is the revised version of a manuscript previously submitted to Nature Microbiology. The authors have made great efforts and the manuscript has greatly improved.

I have only some comments to give on this revised version

1. In the title, abstract, result section and in the headline of figure legends, the authors mention enhanced biofilm formation. To avoid misunderstanding that needs to be changed to enhanced 3D biofilm formation.
2. In the extended Figure 10 the authors interpret the peak and drop during the first hour of phage infection as susceptibility of bacteria to the phage. Is the infection process that fast (within 15 min) considering that phage absorption upon infection initiation can already take much longer. What about the (slight) drop in OD after 2 h?
3. It might be informative to add a scheme with the peptidoglycan fragments created by the different enzymatic digests.

Reviewer #2 (Remarks to the Author):

Nothing to add

Reviewer #3 (Remarks to the Author):

In the work "Bacteria use exogenous peptidoglycan (PG) a danger signal to trigger biofilm formation" the authors show that exposure of bacteria to peptidoglycan leads to broad changes in gene expression resulting in taller, uniquely structured biofilms. Additionally the authors show that the source of this exogenous peptidoglycan can be from other bacterial species or kin that have undergone lytic phage infection. The authors have gone above-and-beyond in characterizing infections by the T7-like vibriophage N4 in their system. Biofilms and phage infections have been extensively studied in *V. cholerae* and this work begins to further show how the biofilm growth state is influenced by and influences interactions with phages.

Figure 1 beautifully shows a changes in biofilm architecture and volume when *V. cholerae* are exposed to the vibriophage N4. Extended exposure of *V. cholerae* to phages similar to N4 usually selects for phage receptor mutants, however in the context of these experiments the bacteria appear to remain susceptible to the phage. Mutation of the receptor(s) can often be deleterious in

the environment of the host and often impact biofilm formation. If the unique biofilm structures induced by peptidoglycan sensing are a protective method that allows bacteria to avoid receptor mutations then it may be worth the authors commenting on. If nothing else, it may be helpful for the authors to speculate as to why they did not observe receptor mutations. Experiments with the *trxA* strain were clever and it is clear a significant amount of effort went into the variety of phage related experiments in this work.

Phages encode diverse lysis cassettes which can include multiple different PG targeting enzymes, including endopeptidases, amidases, and lytic transglycosylases. Figure 3 shows that digestion by such enzymes reduces the observed biofilm phenotype upon exposure to PG, thereby suggesting that phages suppress this signaling even as they induce it. I was wondering if the authors had tested whether phages with disparate lysis cassettes still induced this biofilm phenotype?

Figure 4 utilizes different reporters to show transcriptional responses to peptidoglycan. I find this visualization to be an excellent way to convey a large amount of data. Changes after PG exposure for both the *c-di-GMP* and *vsp-I* reporter appear to occur on the timescale of hours and are spatially distributed throughout the biofilm. The infection cycle of the vibriophage N4 on *V. cholerae* C6706 in these conditions occurs in minutes and *c-di-GMP* signaling is often used for rapid responses to environmental changes. Do the authors have any insight on to how their reporter data temporally fits within this system? Additionally either phage or peptidoglycan exposure should primarily occur on the perimeter of the biofilm, I was wondering if the authors could speak to how the signal would be propagated throughout the biofilm or if the changes observed are primarily due to density dependent changes in expression of the reporters.

It is interesting that the anti-phage CBASS system, *CapV-DncV* appears upregulated upon exposure to peptidoglycan. Do the authors think that this autolysis pathway could be involved in the restricting of biofilms after sensing peptidoglycan? Moreover, these genes, as well as many of the differentially regulated genes fall under regulation of quorum sensing genes such as *HapR* which itself appears to be differentially regulated by peptidoglycan exposure. The authors rule out cell density as an initiator of their biofilm phenotype, but do not comment on how the altered architecture may itself alter localized cell density and alteration of the quorum state of biofilm cells.

This phenomenon occurs throughout gram negative and gram-positive bacteria suggesting a conserved, or convergently evolved mechanism for exogenous peptidoglycan sensing and signal transduction. It makes sense that bacteria will have evolved a mechanism to generally use peptidoglycan as a "danger signal" as many different threats, such as the host, environmental changes, or antibiotics can cause lysis and release of peptidoglycan. Phage generally only infect a limited host range of closely related bacteria. Therefore in a complex environment, sensing a such broad signal might not be relevant to a nearby unrelated and non-susceptible bacterium. I was wondering if the authors could speak to why they believe phages to be a primary source for this signal and the environmental context where such a biofilm response to phage infection may prove beneficial.

Reviewer #3 (Remarks on code availability):

The code was readily available with clear installation instructions and a practice dataset. However, I did not run the code on the practice data so I cannot speak to how smoothly it runs.

Version 2:

Decision Letter:

13th November 2024

Dear Knut,

I am pleased to accept your Article "Bacteria use exogenous peptidoglycan as a danger signal to trigger biofilm formation" for publication in *Nature Microbiology*. Thank you for having chosen to submit your work to us and many congratulations.

Over the next few weeks, your paper will be copyedited to ensure that it conforms to *Nature Microbiology* style. We look particularly carefully at the titles of all papers to ensure that they are relatively brief and understandable.

Once your paper is typeset, you will receive an email with a link to choose the appropriate publishing options for your paper and our Author Services team will be in touch regarding any additional information that may be required. Once your paper has been scheduled for online publication, the *Nature* press office will be in touch to confirm the details.

Please note that *Nature Microbiology* is a Transformative Journal (TJ). Authors may publish their research with us through the traditional subscription access route or make their paper immediately open access through payment of an article-processing charge (APC). Authors will not be required to make a final decision about access to their article until it has been accepted. [Find out more about Transformative Journals](https://www.springernature.com/gp/open-research/transformative-journals)

Authors may need to take specific actions to achieve [compliance](https://www.springernature.com/gp/open-research/funding/policy-compliance-faqs) with funder and institutional open access mandates. If your research is supported by a funder that requires immediate open access (e.g. according to [Plan S principles](https://www.springernature.com/gp/open-research/plan-s-compliance)) then you should select the gold OA route, and we will direct you to the compliant route where possible. For authors selecting the subscription publication route, the journal's standard licensing terms will need to be accepted, including [self-archiving policies](https://www.nature.com/nature-portfolio/editorial-policies/self-archiving-and-license-to-publish). Those licensing terms will supersede any other terms that the author or any third party may assert apply to any version of the manuscript.

With kind regards,

P.S. Click on the following link if you would like to recommend Nature Microbiology to your librarian <http://www.nature.com/subscriptions/recommend.html#forms>

** Visit the Springer Nature Editorial and Publishing website at http://editorial-jobs.springernature.com?utm_source=ejP_NMicro_email&utm_medium=ejP_NMicro_email&utm_campaign=ejP_NMicro for more information about our career opportunities. If you have any questions please click [here](mailto:editorial.publishing.jobs@springernature.com).

Response to Reviewer #1:

Reviewer #1: In this work, the authors observed that the short-term exposure of biofilms of the susceptible *Vibrio cholerae* El Tor strain C6706 to the lytic phage N4 caused the formation of elevated structured three dimensional biofilms with a distinct transcriptional profile and elevated concentration of the second messenger cyclic di-GMP compared to otherwise flat unstructured biofilm formation when grown in LB medium at body temperature. Subsequently, the compound(s) derived from cell lysate that trigger elevated biofilm formation were identified as peptidoglycan. Peptidoglycan was subsequently identified as a broad stimulator of biofilm formation in Gram-negative and Gram-positive bacteria equally as peptidoglycan from different species was able to induce elevated biofilm formation.

This work describes a novel twist in the delicate interplay between a lytic phage and biofilm-forming bacteria with the identification of the components that direct the interactions. The work has been well performed. I am wondering, however, what arguments/interpretation do the authors have with respect to the observed elevated biofilm formation upon phage exposure in rich medium at body temperature? Was a similar phenomenon also observed at lower temperature and/or using minimal medium?

Author response: We are grateful for the reviewer's positive evaluation of our manuscript, and for the constructive requests for clarifications.

Regarding the Reviewer's question about growth media and incubation temperatures:

At the very beginning of our study, we explored different temperatures and media for investigating the interactions of Vibriophage N4 and *V. cholerae*. We found that Vibriophage N4 displays less infection at lower temperatures and in media with lower complexity. The strongest infection was observed in rich medium (LB) at 37 °C, which is why we focused on this condition for our study. We now show the phage infection dynamics in different media and different temperatures in the new Extended Data Figure 1, copied below for convenience. We also mention these results in the revised manuscript in line 66 (changes in the manuscript document are marked in yellow).

Regarding the Reviewer's question about whether a similar phenomenon was observed at lower temperature and/or in minimal medium:

In response to this question, we performed additional experiments to test if biofilm formation in response to peptidoglycan exposure is also observed in different temperatures and in different media. The results are shown in the new Extended Data Figure 8 (copied below for convenience). These results illustrate that in all three media we used (M9 minimal medium, tryptone broth, and LB) and in all temperatures we used (28 °C and 37 °C), peptidoglycan exposure induced the formation of 3D biofilms. Consequently, this key finding of our study is a robust phenomenon that does not rely on a narrow range of the growth conditions. We mention these results in the revised manuscript in lines 181-182.

Extended Data Figure 8: Exogenous peptidoglycan induces biofilm formation of *V. cholerae* in different media and temperatures. *V. cholerae* biofilm growth in our microfluidic system was measured in the presence (+) or absence (-) of 300 μg/mL exogenous peptidoglycan (PG), at a particular temperature (28 °C or 37 °C) and in a particular liquid medium (M9 minimal medium with 0.5% glucose, TB, LB). As growth rates strongly differ between different media and temperatures, we measured the 3D biofilm biovolume fraction with height $H > 3 \mu\text{m}$ in a given growth condition at the time of maximum biofilm height, which is a time that differed between different growth conditions. The time at which the maximum biovolume fraction at heights $H > 3 \mu\text{m}$ occurs is as follows: for M9, $t = 7.5\text{-}13.5 \text{ h}$ at 28 °C and 9-11 h at 37 °C; for TB, $t = 3.5\text{-}4.5 \text{ h}$ at 28 °C and 2.5-5.0 h at 37 °C; for LB, $t = 2.5\text{-}4.5 \text{ h}$ at 28 °C and 2-3 h at 37 °C). In each growth condition, the biofilm biovolume was measured at the same time for the PG-exposed and unexposed condition. In all growth conditions, PG-exposure resulted in a significant enhancement of 3D biofilm formation. Bars indicate the mean of $n = 3$ biological replicates, error bars indicate the standard deviation and individual data points are shown. Statistical significances were calculated using a Student's t -test; * = $p < 0.1$; ** = $p < 0.01$; *** = $p < 0.001$.

General comments

Introduction: Should contain relevant information about *Vibrio cholerae* relevant phages and/or phage/biofilm interactions.

We agree that more context on the interactions of phages and *V. cholerae* would be helpful, yet we had to cut ~200 words from the manuscript due to length restrictions imposed by the journal. Nevertheless, in response to the reviewer comment, we added statements and references to the introduction regarding the interaction of phages and *V. cholerae*, and regarding phage-biofilm interactions. These new sentences are in lines 50-57.

I.173: It seems that the peptidoglycan fragments that induce biofilm formation are quite large and thereby unique in their capability to conduct a biological reaction. Were these compounds digested with peptidoglycan processing enzymes other than lysozyme? Such a treatment should in any case abolish the biofilm inducing capability if the length is indeed a determinative factor.

We appreciate the reviewer's suggestion. To narrow down the structural nature of the biofilm-inducing PG components we have now performed additional experiments using distinct peptidoglycan degrading enzymes: lysozyme, lytic transglycosylase, endopeptidase, amidase. The resulting data are presented in the new Fig. 3f (copied for convenience below) and described in the manuscript in lines 186-195 (copied here for convenience):

*Digestion of purified *V. cholerae* peptidoglycan with lysozyme or amidase resulted in a loss of the biofilm induction capacity, which*

confirms that exposure to exogenous peptidoglycan is sufficient for inducing 3D biofilm formation in *V. cholerae*. Digests of peptidoglycan with endopeptidase or lytic transglycosylase retain some biofilm induction capacity, but with a significant reduction compared to undigested peptidoglycan (Fig. 3f). Considering the most abundant muropeptides produced in both the lytic transglycosylase and endopeptidase digestions³⁰, we speculate that tetrapeptide anhydro-disaccharides (either free or as part of uncrosslinked peptidoglycan chains) may be the components of peptidoglycan that cause biofilm induction.

Caption of Figure 3f: *V. cholerae* WT cells were exposed to purified *V. cholerae* PG (300 μg mL⁻¹ in LB) which was either undigested, or treated with enzymes that cleave specific bonds in PG.

I.185: Any speculations about the nature of alternative compound(s) that trigger elevated biofilm formation in spheroplasts?

We are of course very interested in uncovering additional compounds from lysed cells that trigger biofilm formation, as it is possible that bacteria employ additional danger signals beyond exogenous peptidoglycan. We are therefore currently studying which compounds trigger biofilm formation in spheroplast lysates. We know that the active compounds are not proteins or nucleic acids, and we know that the active compounds are larger than 3 kDa (see Fig. 3b and Extended Data Fig. 7). However, the identification of the exact compounds is a multi-year research project. We therefore did not add speculative statements in our manuscript regarding the identity of such compounds. Near the end of the revision of our manuscript, a publication has reported that the polyamine norspermidine (molecular weight: 131 Da) can induce biofilm formation (PMID: 38443393), and we added a reference to this recent publication in the discussion (lines 294-295). However, this molecule is unlikely to induce biofilm formation in our system, as we know that the active compound(s) in our system are larger than 3 kDa (see Fig. 3b).

I.291: Not sure that this is directly comparable as the peptidoglycan fragments recognized by NOD1 and NOD2 receptors are considered to be the subunits of peptidoglycan. There should be some speculation about the unique nature of the signal as well as the signaling mechanism.

We modified the relevant statement in the *Discussion* section which mentions NOD1 and NOD2, to highlight that these receptors recognize specific subunits of peptidoglycan (now lines 286-289).

The new experiments we performed for the revision using different peptidoglycan digests provide further information about the peptidoglycan signal (resulting in our new Fig. 3f and manuscript lines 186-195, as described in response to the above question by the reviewer).

Regarding the reviewer's question about how the exogenous peptidoglycan is sensed: We have tested the impact of single-gene deletions of peptidoglycan processing enzymes in *V. cholerae*, and all of them showed a substantial reduction in biofilm induction capacity, which did not allow us to draw meaningful conclusions. We therefore did not include these non-conclusive data in our manuscript. We comment on the sensing mechanism in lines 239-241.

Extended Data Figure 1c: This Figure is not clear with bacterial colonies poorly recognized, if at all.

[The relevant figure has been renamed *Extended Data Figure 2* in the revised manuscript.]

We agree that the photographs of the colony morphology were not easy to interpret. As we do not have a suitable apparatus for imaging the colony morphology with sufficiently high contrast and high resolution, we cannot acquire better photos, and we have consequently chosen to remove the photographs of the colony morphology from this figure. The crystal violet assay that was performed with the cells harvested from the phage-exposed biofilms yields the same conclusion as the photographs of the colony morphology (i.e. that the cells in the phage-exposed biofilms are not matrix hyper-producing mutants). Therefore, we now only show the results from the crystal violet assay in Extended Data Figure 2c.

Response to Reviewer #2:

Reviewer #2: In this very interesting manuscript, the authors propose a new mechanism for bacterial community sensing of damage to neighboring cells. Bacteria exposed to phages, which cause cell lysis, or to cell lysates formed 3D biofilms, as opposed to unexposed bacteria, which primarily grew in a 2D monolayer. Bacteria in biofilms were found to be resistant to phage predation. The same phenomenon was observed in bacteria exposed to peptidoglycan alone, suggesting it is the component being sensed. If peptidoglycan was damaged by lysozyme this phenomenon was no longer observed. Exposure to peptidoglycan induced changes in gene expression that included increases in c-di-GMP, an important bacterial second messenger, and increases in biofilm matrix production, which makes sense given the obvious changes in biofilm production on a larger scale. Remarkably, this effect of increased biofilm formation in response to peptidoglycan exposure was observed in several bacterial species and did not require peptidoglycan from any particular species, suggesting it is a very common mechanism for protecting a community.

The manuscript is very well written. I have only minor suggestions for improvement.

Specific points:

1) Congratulations to the authors for very interesting and well-described work!

Author response: We are grateful to the reviewer for the positive evaluation of our manuscript, and for the constructive suggestions for improvement, which we have addressed point-by-point below.

2) Fig 4: it would be beneficial to highlight which virulence genes are upregulated in response to peptidoglycan as this could be a major factor during an infection (appears to be TCP production and MakA toxin. It is curious that *TcpA* is apparently not upregulated but other parts of the operon are.)

This is a very good suggestion. Our transcriptome data indeed show that the *tcp* operon and the *mak* operon are upregulated. Several other virulence factors are also upregulated, such as genes coding for the hemolysin HlyA and the protease HapA. We now mention this in the main manuscript, lines 216-217 (changes in the manuscript document are marked in yellow). We now also highlight this aspect in the conclusion paragraph, line 316.

The reviewer also commented on the fact that *tcpA* is not listed in our list of upregulated genes (Supplementary Table 1), even though the other genes of the *tcp* operon are listed in this table. This is due to the fact that in our dataset *tcpA* has a fold-change of 1.82 (false-discovery-rate adjusted *p*-value < 0.001), which is just below our cut-off of >2-fold upregulation for the list of genes in Supplementary Table 1. Another gene from the *tcp* operon is also just below the >2-fold cut-off: *tcpE*, which is upregulated in our dataset with a fold change of 1.91 (false-discovery-rate adjusted *p*-value < 0.001). The other genes in the *tcp* operon display an upregulation that is >2-fold, and are consequently listed in Supplementary Table 1.

3) Fig 5: Are biofilms formed by another mechanism also resistant to phage predation or is this specific to peptidoglycan sensing? This control would add to the conclusions.

To answer this question, we performed additional experiments. In these experiments, we used a strain into which we introduced the amino acid substitution W240R in the VpvC protein, which leads to elevated c-di-GMP levels, which ultimately causes biofilm matrix hyper-production, and a “rugose” (wrinkly) colony morphology on agar plates. Except for this amino acid substitution, this rugose strain is identical to the *V. cholerae* strain that was used for the key experiments in Figures 1-4. Whenever a cell of this rugose strain attaches to a glass surface, the cell grows into a 3D biofilm colony without requiring exposure to peptidoglycan. Our lab has characterized 3D biofilm formation dynamics of such rugose strains extensively (PMID: 38511867, 36288405, 31156716).

Using this rugose strain, we grew biofilms in our flow chamber device with LB medium for 3 h (so that 3D biofilm colonies were present), prior to exposure to purified phages in LB (or the control treatment: LB without phages). We then monitored the biofilm biovolume in the presence/absence of phages using confocal microscopy for 4 h. These experiments with the rugose strain revealed that in the presence of phages, the biofilms continued to grow almost as fast as in the absence of phages. Even though some limited phage infection may have occurred, these biofilm growth curves indicate that the overall biofilm population was protected from phage attack. We also performed a control experiment in which we performed phage infection assays with the rugose strain in shaking liquid cultures (i.e. not in biofilm conditions). These control experiments showed that the rugose strain is not intrinsically resistant to phages (see the drop in OD₆₀₀ in the first hour after phage infection).

Taken together, these two experiments indicate that 3D biofilm formation of the rugose strain (which is independent of exposure to exogenous peptidoglycan) leads to a protection from phage predation. We have included these data as the new Extended Data Fig. 10 (copied below for convenience), and we describe these findings in the manuscript in lines 250-253.

Response to Reviewers:

Reviewer #1 (Remarks to the Author):

This is the revised version of a manuscript previously submitted to Nature Microbiology. The authors have made great efforts and the manuscript has greatly improved. I have only some comments to give on this revised version

1. In the title, abstract, result section and in the headline of figure legends, the authors mention enhanced biofilm formation. To avoid misunderstanding that needs to be changed to enhanced 3D biofilm formation.

Author response: The current title “Bacteria use exogenous peptidoglycan as a danger signal to trigger biofilm formation” has 84 characters including spaces, which is just below the limit of 90 characters for Nature Microbiology. Expanding the title to “...trigger three-dimensional biofilm formation...” would break this limit. We therefore did not modify the title.

However, as suggested by the reviewer, we added the term “three-dimensional” in the abstract, lines 31, 32, 37. We also added “3D” to the figure titles for Fig. 2 and Fig. 3, ED Fig. 2, ED Fig. 6, ED Fig. 7, ED Fig. 8.

2. In the extended Figure 10 the authors interpret the peak and drop during the first hour of phage infection as susceptibility of bacteria to the phage. Is the infection process that fast (within 15 min) considering that phage absorption upon infection initiation can already take much longer. What about the (slight) drop in OD after 2 h?

Phage adsorption and infection of the wild type strain in shaking liquid culture occurs within a few minutes (Extended Data Fig. 4a), and the release of progeny phages occurs within 16 minutes (Extended Data Fig. 4a). It is reasonable to assume that under the same conditions (which are used in Extended Data Fig. 10b), a similar duration of phage infection would be observed in the rugose strain – which is identical to the wild type except for an amino acid exchange in the *vpvC* gene (*vpvC^{W240R}*) that results in elevated c-di-GMP levels.

We added the reference to Extended Data Fig. 4a to the caption of Extended Data Fig. 10b, to clarify the issue raised by the reviewer.

Regarding the slight drop in OD after 2 h in Extended Data Fig. 10b: During incubation under shaking conditions, the rugose strain forms aggregates even without peptidoglycan sensing. It is possible that the slight drop in OD after 2 h indicates a second wave of phage infection, for which the spread of phage infection through the population is slowed down by the presence of the aggregates – we now also added a comment regarding this possibility in the caption of Extended Data Fig. 10b.

3. It might be informative to add a scheme with the peptidoglycan fragments created by the different enzymatic digests.

We have now generated a schematic diagram that indicates the cleavage sites of the different enzymes that were used. This scheme has been added to Fig. 3f, next to the bar graph that reports data from the different enzymatic digests.

Reviewer #2 (Remarks to the Author):

Nothing to add

Reviewer #3 (Remarks to the Author):

In the work “Bacteria use exogenous peptidoglycan (PG) a danger signal to trigger biofilm formation” the authors show that exposure of bacteria to peptidoglycan leads to broad changes in gene expression resulting in taller, uniquely structured biofilms. Additionally the authors show that the source of this exogenous peptidoglycan can be from other bacterial species or kin that have undergone lytic phage infection. The authors have gone above-and-beyond in characterizing infections by the T7-like vibriophage N4 in their system. Biofilms and phage infections have been extensively studied in *V. cholerae* and this work begins to further show how the biofilm growth state is influenced by and influences interactions with phages.

Author response: We are grateful that the reviewer appreciates the manuscript, and the novelty of the results.

Figure 1 beautifully shows a changes in biofilm architecture and volume when *V. cholerae* are exposed to the vibriophage N4. Extended exposure of *V. cholerae* to phages similar to N4 usually selects for phage receptor mutants, however in the context of these experiments the bacteria appear to remain susceptible to the phage. Mutation of the receptor(s) can often be deleterious in the environment of the host and often impact biofilm formation. If the unique biofilm structures induced by peptidoglycan sensing are a protective method that allows bacteria to avoid receptor mutations then it may be worth the authors commenting on. If nothing else, it may be helpful for the authors to speculate as to why they did not observe receptor mutations.

During the brief exposure period to phages (only 8 hours), our control experiments in Extended Data Figure 2 did not show that there was a significant population of mutants that displayed altered biofilm formation or altered phage susceptibility in liquid culture. However, as pointed out by the reviewer, it is expected that for long durations of phage exposure, phage receptor mutations or biofilm hyper-producing mutants would become abundant in the population. We mention in the manuscript that we speculate that we don't see these populations in our experiments as we only look at relatively brief periods of phage exposure (up to 8 hours) – in lines 85-98. We also added text to the introduction that highlights the different types of adaptation that are expected to occur on different time scales: mutation vs. regulation (lines 45-49).

The biofilms that are formed in response to peptidoglycan sensing do protect against phages (Fig. 5). Therefore, phage-sensitive cells can proliferate in the protected environment of a biofilm, circumventing the need to mutate phage receptors for survival (which often results in a fitness cost such as reduced growth rate). As requested by the reviewer, we now added a comment about this in the relevant section (lines 261-262).

Experiments with the *trxA* strain were clever and it is clear a significant amount of effort went into the variety of phage related experiments in this work.

We are grateful that the reviewer values the effort that went into our experiments!

Phages encode diverse lysis cassettes which can include multiple different PG targeting enzymes, including endopeptidases, amidases, and lytic transglycosylases. Figure 3 shows that digestion by such enzymes reduces the observed biofilm phenotype upon exposure to PG, thereby suggesting that phages suppress this signaling even as they induce it. I was wondering if the authors had tested whether phages with disparate lysis cassettes still induced this biofilm phenotype?

We did not investigate different phage strains in this study after we realized (in Fig. 2) that the signal is not phage-specific, but instead a general danger signal that can be released in a phage-independent manner. Studying if phages with different lysis cassettes are able to inhibit the effect of the PG signal by degrading the PG would be an interesting avenue for future investigations!

Figure 4 utilizes different reporters to show transcriptional responses to peptidoglycan. I find this visualization to be an excellent way to convey a large amount of data. Changes after PG exposure for both the c-di-GMP and vsp-I reporter appear to occur on the timescale of hours and are spatially distributed throughout the biofilm. The infection cycle of the vibriophage N4 on *V. cholerae* C6706 in these conditions occurs in minutes and c-di-GMP signaling is often used for rapid responses to environmental changes. Do the authors have any insight on to how their reporter data temporally fits within this system? Additionally either phage or peptidoglycan exposure should primarily occur on the perimeter of the biofilm, I was wondering if the authors could speak to how the signal would be propagated throughout the biofilm or if the changes observed are primarily due to density dependent changes in expression of the reporters.

Regarding the reviewer's question about the timing of the fluorescent-reporter data, and how it fits into the transcriptome data: Our results from the transcriptomes of cells exposed to peptidoglycan for only 10 min do indeed show that there is a rapid transcriptional response, which suggests that changes in c-di-GMP levels regulate the biofilm response (i.e., matrix production). However, for the fluorescent-reporter data in Fig. 4c,d, we observe substantial differences after 1-2 hours. This delay is likely due to two effects:

1. The dynamic range of these fluorescent reporters (shown for the c-di-GMP reporter in Extended Data Fig. 9a) is substantially lower than for RNA-seq based transcriptome measurements. We therefore expect to detect even small significant differences quickly after PG-exposure with the RNA-seq measurements, and that the fluorescent reporters only show substantial differences later on.
2. Fig. 3g shows that exposing cells to peptidoglycan for a period of only 5 min induces the same level of 3D biofilm formation as exposure for 180 min. This finding for brief peptidoglycan exposure periods suggests that peptidoglycan exposure results in a phenotypic switch that induces a biofilm formation program, which is passed on to the daughter cells that have never directly been in contact with peptidoglycan. This is now highlighted in lines 207-210. The high *vps*-expression and elevated c-di-GMP levels that are detected in Fig 4c,d are therefore probably not the result of direct PG-sensing, but of the PG-induced switch to the biofilm formation program, which is propagated to daughter cells during biofilm formation.

Regarding the reviewer question about peptidoglycan propagation inside the biofilm: This is an interesting question, but it is unfortunately not possible to measure this, because no suitable techniques are available for such measurements.

It is interesting that the anti-phage CBASS system, CapV-DncV appears upregulated upon exposure to peptidoglycan. Do the authors think that this autolysis pathway could be involved in the restricting of biofilms after sensing peptidoglycan? Moreover, these genes, as well as many of the differentially regulated genes fall under regulation of quorum sensing genes such as HapR which itself appears to be differentially regulated by peptidoglycan exposure. The authors rule out cell density as an initiator of their biofilm phenotype, but do not comment on how the altered architecture may itself alter localized cell density and alteration of the quorum state of biofilm cells.

Regarding the reviewer's comment about the CBASS system: Indeed, the CBASS phage defense system genes *capV*, *dncV*, *cap2*, and *cap3* are all substantially upregulated following 10 min of peptidoglycan exposure. However, the 10 other phage defense systems that are detectable in the genome of our *V. cholerae* strain using the phage defense finder online tool (<https://defensefinder.mdmlab.fr/>) are not significantly upregulated. It is possible that the upregulation of the CBASS system contributes to the phage tolerance of cells that reside in biofilms. However, even biofilms that formed in a peptidoglycan-independent manner are phage-tolerant (Extended Data Fig. 10) – yet we don't know if in these biofilms CBASS is also

upregulated. We added a note regarding the upregulation of the CBASS system following peptidoglycan-exposure to the manuscript in lines 224-225.

Regarding the reviewer's comment about quorum sensing regulation: It was indeed surprising that *hapR* was upregulated during peptidoglycan exposure. The transcription factor HapR is usually associated with repression of biofilm formation in *V. cholerae* (DOI: 10.1046/j.1365-2958.2003.03688.x). However, quorum-sensing independent regulation of biofilm matrix genes in *V. cholerae* is possible: For example, a study by Waters et al. (DOI: 10.1128/JB.01756-07) has shown that c-di-GMP activates VpsT and VpsR, which subsequently upregulates the transcription of biofilm matrix genes. Our transcriptome data in Fig. 4 suggests that biofilm formation of *V. cholerae* as a response to peptidoglycan exposure could be induced via a pathway that circumvents the HapR-mediated repression of *vps* genes. In the presence of high levels of c-di-GMP, VpsT- and VpsR-mediated biofilm matrix upregulation has been shown to take precedence over transcriptional repression by HapR. However, these hypotheses and the quorum sensing state of cells in different regions of the biofilm require further investigation, to disentangle the regulatory circuitry in play during peptidoglycan sensing.

This phenomenon occurs throughout gram negative and gram-positive bacteria suggesting a conserved, or convergently evolved mechanism for exogenous peptidoglycan sensing and signal transduction. It makes sense that bacteria will have evolved a mechanism to generally use peptidoglycan as a “danger signal” as many different threats, such as the host, environmental changes, or antibiotics can cause lysis and release of peptidoglycan. Phage generally only infect a limited host range of closely related bacteria. Therefore in a complex environment, sensing a such broad signal might not be relevant to a nearby unrelated and non-susceptible bacterium. I was wondering if the authors could speak to why they believe phages to be a primary source for this signal and the environmental context where such a biofilm response to phage infection may prove beneficial.

Our study demonstrates that exogenous peptidoglycan is used by different species as a signal that induces biofilm formation. However, this process is independent of how the exogenous peptidoglycan is produced: by phages or any other lysis-inducing condition. We therefore do not claim in the manuscript that phage-induced lysis is the primary source of this signal – we serendipitously discovered this signal in the context of phage predation.

To clarify this point, we fine-tuned the second paragraph of the Discussion section (lines 317-329) in the manuscript, to highlight that exogenous peptidoglycan is not just a signal resulting from phage-induced lysis, but other conditions also lead to the release of this signal. We also mention in this paragraph that the biofilm state protects cells against other biotic and abiotic stresses beyond phage predation.

Reviewer #3 (Remarks on code availability):

The code was readily available with clear installation instructions and a practice dataset. However, I did not run the code on the practice data so I cannot speak to how smoothly it runs.

We double-checked that the downloadable code indeed works on an independent computer.